# Hardware implementation of Bayesian network based on two-dimensional memtransistors

Yikai Zheng[1], Harikrishnan Ravichandran[1], Thomas F. Schranghamer[1], Nicholas Trainor[2,3], Joan M. Redwing[2,3] & Saptarshi Das [1,2,3,4] ✉

Bayesian networks (BNs) find widespread application in many real-world probabilistic problems including diagnostics, forecasting, computer vision, etc. The basic computing primitive for BNs is a stochastic bit (s-bit) generator that can control the probability of obtaining '1' in a binary bit-stream. While silicon-based complementary metal-oxide-semiconductor (CMOS) technology can be used for hardware implementation of BNs, the lack of inherent stochasticity makes it area and energy inefficient. On the other hand, memristors and spintronic devices offer inherent stochasticity but lack computing ability beyond simple vector matrix multiplication due to their two-terminal nature and rely on extensive CMOS peripherals for BN implementation, which limits area and energy efficiency. Here, we circumvent these challenges by introducing a hardware platform based on 2D memtransistors. First, we experimentally demonstrate a low-power and compact s-bit generator circuit that exploits cycle-to-cycle fluctuation in the post-programmed conductance state of 2D memtransistors. Next, the s-bit generators are monolithically integrated with 2D memtransistor-based logic gates to implement BNs. Our findings highlight the potential for 2D memtransistor-based integrated circuits for non-von Neumann computing applications.

The concept of a Bayesian network (BN) is deep rooted within natural intelligence. Animals gather information from their surroundings with the help of their sensory organs and process this information using their brain to make decisions, enabling their survival. However, gathering accurate information is often very difficult in practice either due to the limitations of sensory organs or due to noisy environment. For example, visual cues are an unreliable source of information for freshwater fish like the rainbow trout to identify the presence of a predator. In contrast, chemical cues released into the water from an injured fish are more reliable indicators of a predatory event[1]. The decision to invoke an alarm response, therefore, depends on how the brain processes the visual and chemical cues based on their relative probability of success from prior experiences. While the neural basis of

such computations is relatively unknown, the mathematical construct is represented using a BN with theoretical foundation in Bayes' theorem.

A BN is a probabilistic graphic network used to estimate and infer the probability of interdependent events[2]. Figure 1a shows the basic building block of a BN, comprising a parent node, $A$, a child node, $B$, and an edge connecting the two. Each node represents an event, e.g., the presence of a chemical cue ($A$) and the presence of a predator ($B$), and the connection represents how two events are mutually dependent. The dependence is provided in a conditional probability table (CPT) which contains the conditional probability (likelihood) values $P(B/A)$ and $P(B/A^c)$, where $A^c$ is the complement of the event $A$. In the present example, these represent the likelihood of the presence of a

[1]Engineering Science and Mechanics, Penn State University, University Park 16802 PA, USA. [2]Materials Science and Engineering, Penn State University, University Park 16802 PA, USA. [3]Materials Research Institute, Penn State University, University Park 16802 PA, USA. [4]Electrical Engineering and Computer Science, Penn State University, University Park 16802 PA, USA. ✉e-mail: sud70@psu.edu

*(a)* <u>*Bayesian Networks (BNs)*</u>        *(b)* <u>*Hardware Implementation of BN*</u>

$$P(B) = P(B/A)P(A) + P(B/A^c)P(A^c)$$
$$P(A^c) = 1 - P(A)$$

$$B = AX_1 + A^c X_2$$

*(c)* <u>*Example Applications of BNs*</u>

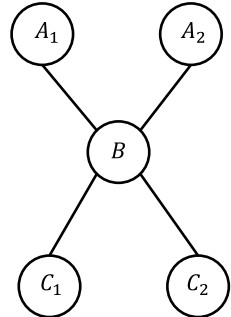

| Events | Ecology | Forecasting | Drug discovery |
|---|---|---|---|
| $A_1$ | Presence of visual cue | Windy day | Drug-1 on gene1 |
| $A_2$ | Presence of chemical cue | Cloudy day | Drug-2 on gene1 |
| $B$ | Presence of predator | Chance of rain | Signaling pathway |
| $C_1$ | Stop swimming | Coffee sale | Antibody production |
| $C_2$ | Stop foraging | Umbrella sale | Hormone secretion |

**Fig. 1 | Bayesian networks (BNs). a** Schematic of the basic building block of a BN, comprising a parent node, A, a child node, B, and an edge connecting the two. Each node represents an event, and the connection represents how two events are mutually dependent. The dependence is provided in a conditional probability table (CPT), which contains the conditional probability (likelihood) values $P(B/A)$ and $P(B/A^c)$, where $A^c$ is the complement of the event. Knowing the probability of occurrence for event A, i.e., $P(A)$, the marginal probability of occurrence of event $B$, i.e., $P(B)$, can be evaluated using Bayes' theorem. **b** Hardware implementation of the 2-node BN in (**a**) using three stochastic bit (s-bit) generators and one 2×1 multiplexer (MUX) circuit. **c** Examples of BN architecture that represent real-life situations from ecology to forecasting and drug discovery, highlighting its usefulness in decision making.

predator when a chemical cue is present ($A$) or absent ($A^c$), respectively. When the probability of occurrence for event A, i.e., $P(A)$, is known, the marginal probability of occurrence of event $B$, i.e., $P(B)$, can be evaluated using Bayes' theorem following Eq. 1.

$$P(B) = P(B/A)P(A) + P(B/A^c)P(A^c) = P(B/A)P(A) + P(B/A^c)[1 - P(A)] \quad (1)$$

$$P(A) + P\left(A^C\right) = 1 \quad (2)$$

In a generic BN, a child node can have multiple parent nodes, and a parent node can have multiple children. For example, Supplementary Fig. S1a shows a BN where the child node, $B$, is connected to 2 parent nodes, $A_1$ and $A_2$. Note that the CPT in this instance contains $N = 4$ entries, which are the conditional probability (likelihood) for the occurrence of event $B$ under all possible combinations of the occurrence of events $A_1$ and $A_2$. Similarly, Supplementary Fig. S1b shows a BN where the parent node, $A$, is connected to 2 children, $B_1$ and $B_2$. In this case, there are 2 CPTs with $N = 2$ entries each.

Note that the probability estimation for a child node requires multiple arithmetic operations such as multiplication, subtraction, and addition. This makes hardware implementation of a BN using conventional silicon complementary metal-oxide-semiconductor (CMOS) technology[3, 4] less attractive because 1) arithmetic operations require circuits consisting of hundreds of transistors, which have large footprints and consume a significant amount of energy,

and 2) the von Neumann bottleneck necessitates storing of the CPT in the memory, which is physically separated from the arithmetic core and therefore requires frequent data shuttling between the two, further aggravating the energy burden. In contrast, even the tiniest brains with very limited numbers of neurons can perform such apparently complex computational tasks with miniscule energy expenditure. The success of biological brains in implementing BNs could lie in the inherently stochastic nature of neural computation.

Drawing inspiration from biology, stochastic computing (SC) has been explored for the hardware implementation of BNs[5]. The key difference from classical computing, where information in presented in the form of binary values (1's and 0's), is that SC encodes information using stochastic bits (s-bits) that are interpreted as probabilities that fall in the interval [0,1]. For instance, the bit-stream $S = [1\,0\,0\,1\,0\,1\,0\,0]$ encodes the value $P(S) = 3/8$, i.e., the probability of finding '1' in the bit-stream $S$. An attractive feature of SC is that arithmetic operations can be performed using simple logic gates[6, 7]. For example, the 2-node BN in Fig. 1a can be realized using a multiplexer (MUX) circuit as shown in Fig. 1b. The output, $B$, of a MUX with two input variables, $X_1$ and $X_2$, and a select line, $A$, is given by Eq. (3).

$$B = AX_1 + A^c X_2 \quad (3)$$

If, instead of being digital variables, $X_1$, $X_2$, and $A$ represent stochastic variables with $P(X_1)$, $P(X_2)$, and $P(A)$ being the respective probability of obtaining '1' in their bit-streams, then $B$ also transforms into a

random variable whose probability is given by Eq. (4).

$$P(B) = P(A)P(X_1) + P(A^c)P(X_2) \qquad (4)$$

Note that, if $P(X_1) = P(B/A)$ and $P(X_2) = P(B/A^c)$, then Eq. (4) transforms into Eq. (1). Therefore, hardware implementation of a child node with a single parent can be accomplished by using 3 s-bit generators and a $2 \times 1$ *MUX*. Interestingly, the *MUX* architecture can be scaled to implement any BN. For example, hardware implementation of the BN in Fig. 1a can be achieved by using 2 s-bit generators to obtain $A_1$ and $A_2$, another 4 s-bit generators to obtain the CPT, and one $4 \times 1$ *MUX* with 2 select lines as shown in Supplementary Fig. S1c. Similarly, Supplementary Fig. 1d shows the hardware architecture for the BN in Supplementary Fig. S1b, consisting of 1 s-bit generator to obtain $A$, another 4 s-bit generators to obtain the 2 CPTs, and 2 $2 \times 1$ *MUX*s.

Note that BN architecture can be used to represent many real-life situations, as shown in Fig. 1c. For example, in the case of the rainbow trout, events $A_1$ and $A_2$ represent the presence of independent visual and chemical cues and event $B$ represents the presence of a predator. Events $C_1$ and $C_2$, meanwhile, represent the decision taken by the rainbow trout to stop swimming and stop foraging, respectively, which are also independent of each other but depend on $B$. Similarly, in forecasting, events $A_1$ and $A_2$ represent the probability of a day being cloudy and windy, respectively, event $B$ represents the probability of rain, and events $C_1$ and $C_2$ may represent the decision to purchase an umbrella or drink coffee, respectively. Finally, a third example is derived from genetics and drug discovery, where events $A_1$ and $A_2$ may represent the probability of expressing *gene* 1 and *gene* 2 when intervening with a specific drug, respectively, event $B$ represents the activation of a critical signaling pathway, and events $C_1$ and $C_2$ represent production of specific hormones or antibodies, respectively. The above discussion exemplifies the usefulness of BNs in depicting causal relationships using acyclic graphs, which can subsequently be used to predict outcomes based on prior knowledge and likelihood. For example, to predict the relative effectiveness between drug-1 and drug-2 that influence expression for *gene* 1 and *gene* 2, respectively, the only experiments that one needs to do is to obtain respective prior results, i.e., $P(A_1)$ and $P(A_2)$. A BN can then be used to obtain marginal likelihoods, i.e., $P(C_1)$ and/or $P(C_2)$, to assess the relative effectiveness of the two drugs.

The fundamental computing primitive for the stochastic computing implementation of a BN is an s-bit generator, which allows control of the output probability of obtaining '1' in a given bit-stream. So far, probabilistic CMOS[8], field-programmable gate arrays (FPGAs)[9–11], memristors[12–14], and spintronic devices[15–21] have been successfully used for BN implementation. However, CMOS- and FPGA-based BN architectures require hundreds of transistors to generate s-bits, which limits their area and energy efficiency[22–27]. In contrast, memristors offer inherent stochasticity in their switching dynamics, which can be exploited to obtain random bits. However, memristor-based BN architectures heavily rely on CMOS peripherals to translate random bits into s-bits and for subsequent logic operations using those s-bits. Recently, spintronic devices such as magnetic random access memory (MRAM)[28] and magnetic tunnel junctions (MTJs)[29–31] have shown potential for BN implementation since s-bits can be obtained by controlling the probability of spin-flip through externally driven current. However, temperature and supply voltage fluctuations can impact the spin-flip probability, which necessitates additional CMOS-based peripheral circuits to remove the bit-bias. In addition, spin-based devices still require CMOS-based logic circuits for BN implementation.

In this work, we demonstrate hardware implementation of a BN using a monolithic memtransistor technology based on two-dimensional (2D) semiconductors such as monolayer $MoS_2$. Memtransistors are three-terminal devices in which the gate terminal allows non-volatile and analog programming of the conductance states, which can then be readout by applying a source-to-drain bias. Our main contributions in this work are 1) the design of an area and energy efficient s-bit generator circuit composed of six memtransistors, allowing it to achieve a tunable probability of obtaining '1' in the bit-stream over the range [0,1], and 2) integration of s-bit generators with a 2D memtransistor-based $2 \times 1$ *MUX* that consists of three *NAND* gates and one *NOT* gate for BN implementation. In brief, we exploit the inherent stochasticity of the charge trapping and detrapping processes in the gate dielectric of the memtransistor as the source of randomness. Our in-memory computing approach based on three-terminal 2D memtransistors not only overcomes the von Neumann limitations of conventional digital CMOS, but also eliminates the need for peripherals, which is inescapable for emerging memristor- and spin-based 2-terminal stochastic devices for BN implementation.

Our choice of monolayer $MoS_2$ is motivated by the fact that atomically thin 2D materials are being considered for advanced technology nodes[32]. It is widely accepted that scaling silicon thickness beyond ~3–4 nm is challenging. Yet, the gate electrostatics demand aggressive reduction in the channel thickness to preserve the desired device performance for sub-10 nm technology nodes[33]. The ultimate channel thickness for a field-effect transistor (FET) would be in the sub-1 nm range, which is difficult to realize using bulk semiconductors[34], making 2D materials a natural choice for ultra-scaled FETs[35–41]. In fact, recent years have witnessed many experimental breakthroughs in the development of high-performance 2D FETs[42–45], neurosynaptic devices[46–50], and very large scale integrated (VLSI) circuits[51–54]. Similarly, theoretical calculations and quantum mechanical simulation have found that the 2D FETs can outperform CMOS HP (high performance) in both energy and delay[55–58].

## Results

### Fabrication and characterization of 2D memtransistors

Figure 2a, b, respectively, show the 2D schematic and optical image of a representative 2D memtransistor based on monolayer $MoS_2$, which is locally back-gated with sputter-deposited 40/30 nm Pt/TiN serving as the back-gate electrode with atomic layer deposition (ALD) grown 50 nm $Al_2O_3$ as the gate dielectric. All back-gate islands were placed on a commercially purchased $SiO_2$/p$^{++}$-Si substrate. As we will discuss later, the analog, non-volatile, and stochastic programming capability offered by the $Al_2O_3$/Pt/TiN gate stack is central to our BN architecture. The monolayer $MoS_2$ used in this work was grown using a metal-organic chemical vapor deposition (MOCVD) technique on a sapphire substrate at 950 °C[45, 59]. Use of an epitaxial substrate and elevated growth temperature ensured a uniform and high quality 2D film, which is critical for the successful demonstration of our BN architecture that involves many 2D memtransistors. For subsequent 2D memtransistor fabrication, the monolayer $MoS_2$ film was transferred from the growth substrate to the $SiO_2$/p$^{++}$-Si substrate with predefined islands of $Al_2O_3$/Pt/TiN. Details on monolayer $MoS_2$ synthesis, film transfer, and fabrication of the local back-gate gate islands, $MoS_2$ memtransistors, and BN architecture can be found in the "Methods" section as well as in the Methods sections of our recent works[45, 60–63].

The film quality and device performance were assessed using optical and electrical measurements. The Raman spectra (Supplementary Fig. S2a) obtained for a representative 2D memtransistor shows two characteristic monolayer $MoS_2$ peaks at 383 cm$^{-1}$ and 404 cm$^{-1}$ corresponding to the in-plane $E_{2g}$ and out-of-plane $A_{1g}$ modes, respectively, with the expected peak separation of ~20 cm$^{-1}$ for monolayer $MoS_2$[64]. Similarly, the photoluminescence (PL) spectra (Supplementary Fig. S2b) shows a peak at 1.83 eV corresponding to the direct bandgap of monolayer $MoS_2$. The transfer characteristics, i.e., source-to-drain current ($I_{DS}$) versus local back-gate voltage ($V_{BG}$), measured using a source-to-drain bias ($V_{DS}$) of 1 V are shown in Fig. 2c in both linear and logarithmic scale for a representative $MoS_2$

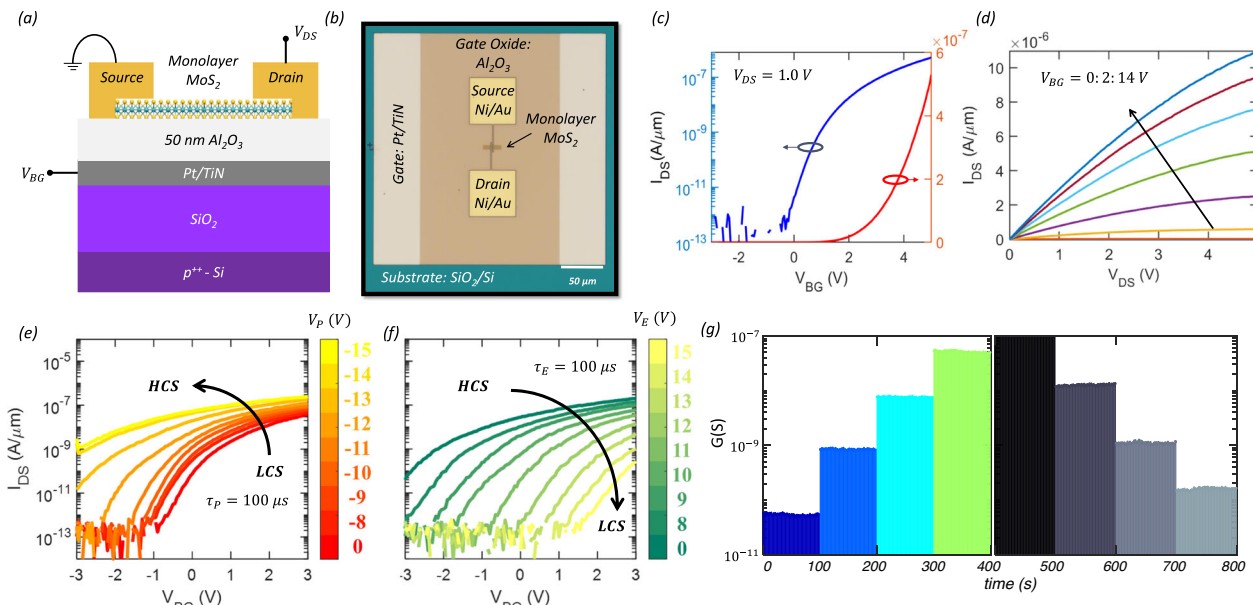

**Fig. 2 | 2D memtransistors. a** 2D schematic and **b** optical image of a representative 2D memtransistor based on monolayer $MoS_2$, which is locally back-gated with sputter-deposited 40/30 nm Pt/TiN serving as the back-gate electrode with atomic layer deposition (ALD) grown 50 nm $Al_2O_3$ as the gate dielectric. All back-gate islands were placed on a commercially purchased $SiO_2/p^{++}$-Si substrate. **c** Transfer characteristics, i.e., source-to-drain current ($I_{DS}$) versus local back-gate voltage ($V_{BG}$), measured at a source-to-drain bias ($V_{DS}$) of 1 V, in linear and logarithmic scale for a representative $MoS_2$ memtransistor with channel length $L = 1\,\mu m$ and channel width $W = 5\,\mu m$. **d** Output characteristics, i.e., $I_{DS}$ versus $V_{DS}$, at different $V_{BG}$ for the same $MoS_2$ memtransistor. **e** Post-programmed and **f** post-erased transfer characteristics of a representative 2D memtransistor after being subjected to negative "Write" ($V_P$) and positive "Erase" ($V_E$) voltage pulses of different amplitudes applied to the local back-gate electrode, each for a duration of $\tau_{P/E} = 100\,\mu s$ **g** Non-volatile retention for 4 representative post-programmed and post-erased conductance states ($G_{MT}$) over 100 s.

memtransistor with a channel length ($L$) of 1 μm and a channel width ($W$) of 5 μm. As expected, n-type transport is observed in $MoS_2$, which is attributed to the pinning of the metal Fermi level near the conduction band[65–67]. Nevertheless, the $MoS_2$ memtransistor exhibits excellent electrostatic gate control with a current on/off ratio ($r_{ON/OFF}$) > $10^5$, a subthreshold slope ($SS$) < 400 mV/decade averaged over 3 orders of magnitude change in $I_{DS}$, minimal gate hysteresis when measured in air, and low gate leakage current. The threshold voltage ($V_{TH}$) was found to be -1.75 V extracted at an iso-current of 10 nA/μm and the electron field effect mobility ($\mu_{FE}$) extracted from the peak transconductance was found to be 5 cm²/V·s. Figure 2d shows the output characteristics, i.e., $I_{DS}$ versus $V_{DS}$, at different $V_{BG}$ for the same representative $MoS_2$ memtransistor. The on-current ($I_{ON}$) reached as high as ~ 11 μA/μm for an inversion carrier density of ~1×$10^{12}$/cm² at $V_{DS} = 5$ V. These results suggest that the monolayer $MoS_2$ film grown using MOCVD is of reasonably good quality, and that the memtransistor fabrication processes including the film transfer are clean and damage-free.

The post-programmed and post-erased transfer characteristics of a representative 2D memtransistor after being subjected to negative "Write" ($V_P$) and positive "Erase" ($V_E$) voltage pulses applied to the local back-gate electrode of varying amplitudes, each for a duration of $\tau_{P/E} = 100\,\mu s$, are shown in Fig. 2e, f, respectively. The negative and positive shift in the respective transfer characteristics can be ascribed to electron trapping and detrapping at and near the $MoS_2/Al_2O_3$ interface, respectively. Note that trap states can originate from defects/imperfections in the dielectric and/or adsorbed species at the 2D/dielectric interface as reported in various earlier studies[68–70]. These states can also be engineered at desired energetic locations by introducing intentional defects in the 2D channel material[51, 71]. Carrier occupancy in these trap states follow Fermi-Dirac distribution. As illustrated using the energy band diagrams in Supplementary Fig. S3, at equilibrium, i.e., in the absence of any gate bias, the trap states with energy levels above the Fermi energy ($E_F$) are empty, whereas the ones

below $E_F$ are filled. When the memtransistor is subjected to a negative "Write" ($V_P$) voltage pulse, electrons are released (detrapped) from these trap states leaving them positively charged. This leads to screening of the back-gate bias, which is reflected as shift in the threshold voltage ($\triangle V_{TH}$). Similarly, when the memtransistor is subjected to a positive "Erase" ($V_E$) voltage pulse, electrons are captured back (trapped) into the trap states, restoring the $V_{TH}$. Note that the number of electrons getting trapped/detrapped can be controlled by both the magnitude and duration of $V_P$ and $V_E$, which allow us to have an analog control of the $\triangle V_{TH}$ and of the conductance state of the memtransistor.

The minimum program/erase pulse width is determined by the trapping/detrapping time constants. Supplementary Fig. S4a–d show the post-programmed and post-erased transfer characteristics of a 2D memtransistor subjected to $V_P$ and $V_E$ voltage pulses of different amplitudes ranging from 8 V to 15 V applied to the local back-gate electrode, each for a duration of $\tau_{P/E} = 100\,\mu s$, 10 μs, 1 μs, and 100 ns, respectively. Clearly, the charge trapping and detrapping processes can occur as fast as 100 ns, which is the limit set by our measurement tools, allowing further improvement in the programming speed[72, 73]. Supplementary Fig. S4e, f show the extracted shift in the threshold voltage ($\triangle V_{TH}$) as a function of $V_{P/E}$ for $\tau_{P/E} = 100\,\mu s$ and $\tau_{P/E} = 100$ ns, respectively. From these results, we can conclude that, for any given pulse magnitude $V_{P/E}$, $\triangle V_{TH}$ becomes smaller as $\tau_{P/E}$ becomes shorter. To retain similar $\triangle V_{TH}$ for smaller $\tau_{P/E}$, larger $V_{P/E}$ is required, which will increase the energy expenditure. Therefore, one needs to strike a balance between fast programmability and energy consumption based on the application needs.

The trapping and detrapping processes were found to be non-volatile, as shown in Fig. 2g for 4 representative post-programmed and post-erased conductance states ($G_{MT}$) over 100 s. We also examined long-term memory retention for the 2D memtransistors and found that states remain distinguishable even after 3 hrs. Memory retention is important to store the CPT and the memtransistors

demonstrate adequate memory performance for the hardware implementation of BNs using SC. The program/erase endurance is also important for the 2D memtransistor. Supplementary Fig. S5 shows the post-programmed and post-erased conductance states of a representative memtransistor, achieved with $V_P = -7$ V and $V_E = 10$ V using $\tau_{P/E} = 100$ ns and measured at $V_{BG} = 0$ V for up to $10^9$ endurance cycles. Clearly, there is no significant change in the two states. While it is desirable to demonstrate endurance for an even higher number of cycles, note that, for the many edge applications, the current endurance results can be sufficient. For example, in weather forecasting, the BN will be used every minute rather than every microsecond; similarly, in medical diagnostics, the BN will be only used several thousand times a day to assess patients.

### Programming stochasticity in 2D memtransistors and design of s-bit generator

Design of hardware for high-quality random bit generation is central to the hardware implementation of BNs. Here, we exploit the cycle-to-cycle variation in the post-programmed and post-erased conductance states ($G_{MT}$) of 2D memtransistors as a source of true randomness. Figure 3a shows the transfer characteristics of a representative MoS$_2$ memtransistor, which is measured each time after applying $V_P = -10$ V and $V_E = 10$ V for $\tau_s = 100$ μs, for a total of 100 cycles and Fig. 3b, c, respectively, show the histograms of post-programmed and post-erased $G_{MT}$ values extracted at $V_{BG} = 0$ V. Clearly, the $G_{MT}$ values follow Gaussian random distributions. The cycle-to-cycle variation in program/erase processes is a direct consequence of the stochastic nature of charge trapping and detrapping observed in most semiconductor/dielectric interfaces[74–76]. In the simple two-state model, a trap state can be electrically neutral or charged, and it can transition between the two states even under equilibrium condition with transition times exponentially distributed. In other words, the state transition dynamics for traps follows the classic Markovian process[77, 78]. In ultra-scaled metal-oxide-semiconductor field effect transistors (MOSFETs) such stochastic state transitions lead to random telegraph noise (RTN). Metastable states are also often involved in the trapping/detrapping processes, making the transition dynamic more complex, rich, and, at the same time, introducing an additional source of randomness[79]. While RTN is not observed in our relatively large area memtransistors, the stochasticity of trapping/detrapping processes manifest during the program/erase operations, thus leading to the cycle-to-cycle variation in $\triangle V_{TH}$.

To translate the stochastic conductance fluctuation into s-bits, we deploy a circuit consisting of six memtransistors ($MT1$, $MT2$, $MT3$, $MT4$, $MT5$, and $MT6$), as shown using the circuit diagram and corresponding optical image in Fig. 3d, e, respectively. The voltage waveforms applied to the nodes N1 and N2, i.e., $V_{N1}$ and $V_{N2}$, respectively, are shown in Fig. 3f. Note that during each clock cycle ($\tau_{clk}$), $V_{N1}$ switches between 0 V, 0 V, and 2 V and $V_{N2}$ switches between $V_P = -7$ V, $V_E = 10$ V, and $V_R = 1$ V. Voltages applied to nodes N3 and N4, i.e., $V_{N3}$ and $V_{N4}$, are held constant at 1 V and 0 V, respectively. This allows programming and erasing of $MT1$ during each $\tau_{clk}$. The voltage readout at node N5, i.e., $V_{N5}$, is shown in Fig. 3g and exhibits stochastic fluctuation. Note that the series connection of memtransistors $MT1$ and $MT2$ represents a voltage divider circuit, and hence $V_{N5}$ is determined by their respective conductance values, i.e., $G_{MT1}$ and $G_{MT2}$. Since $G_{MT1}$ fluctuates from cycle-to-cycle owing to the programming and erasing voltages applied to its local backgate terminal, i.e., N2, so does $V_{N5}$. In other words, the voltage divider translates conductance fluctuations into voltage fluctuations. Figure 3h shows the histogram of $V_{N5}$, which, as expected, follows a random Gaussian distribution with a mean ($\mu_{VN5}$) of 0.40 V and standard deviation ($\sigma_{VN5}$) of 0.02 V.

Next, the Gaussian distribution is broadened by using an inverting amplifier constructed using $MT3$ and $MT4$. Note that the local back-

gate of $MT3$ is shorted to its source at node $N_6$. This ensures that $MT3$ operates as a depletion mode (normally on) transistor or as a load resistor. Figure 3i shows the output, $V_{N6}$, as a function of the input, $V_{N5}$. The slope of the curve is referred to as the gain of the amplifier, and the higher the gain, the wider the broadening of the Gaussian. We achieved a gain of ~24. The gain can be increased further by cascading multiple amplifiers; however, this adds area and energy overhead. Figure 3j shows $V_{N6}$ corresponding to $V_{N5}$ obtained in Fig. 3g. Clearly, the histogram of $V_{N6}$ shown in Fig. 3k exhibits a Gaussian distribution with a mean ($\mu_{VN6}$) of 0.99 V and an increased standard deviation ($\sigma_{VN6}$) of 0.41 V.

Finally, to transform the analog fluctuations seen in $V_{N6}$ into s-bits, a thresholding inverter with a programmable inversion threshold, $V_{IT}$, is constructed using $MT5$ and $MT6$. Figure 3l shows the output, $V_{N7}$, as a function of the input, $V_{N6}$, for different $V_{IT}$. Note that $V_{IT}$ is the magnitude of $V_{N6}$ for which $V_{N7}$ reaches $V_{DD}/2$, i.e., 1 V in the present case. The programmability of $V_{IT}$ is a critical feature that distinguishes 2D memtransistor-based inverters from conventional CMOS-based inverters and allows us to seamlessly obtain the s-bits. Figure 3m shows $V_{N7}$ corresponding to $V_{N6}$ obtained in Fig. 3j for different $V_{IT}$ and Fig. 3n shows the corresponding probability of obtaining '1' in the bit-stream, i.e., $p_s$ as a function of $V_{IT}$. As expected, if $V_{IT}$ is too low, then almost all $V_{N6}$ values translate into $V_{N7} \approx 0$ V, which is reflected as near zero $p_s$. Similarly, if $V_{IT}$ is too high, then almost all $V_{N6}$ values translate into $V_{N7} \approx 2$ V, leading to $p_s = 1$. Between these two extremes, $p_s$ increases monotonically with $V_{IT}$. This clearly shows that we are able to convert the cycle-to-cycle random conductance fluctuations in 2D memtransistor into s-bits with reconfigurable $p_s$ values that lie between [0,1] using the described circuit.

Note that the cycle-to-cycle variation in the programming of 2D memtransistors will lead to fluctuations in the threshold voltage ($V_{TH}$) of $MT6$ and hence in $V_{IT}$ of the thresholding inverter and $p_s$ for the s-bit-stream. Supplementary Fig. S6a-b, respectively, show the distribution of $V_{TH}$ and $V_{IT}$ when $MT6$ is subjected to 50 program/erase/read cycles with $V_P = -7$ V, $V_E = 10$ V, and $\tau_{P/E} = 100$ μs. The means and standard deviations were found to be −0.04 V and 0.08 V for $V_{TH}$, respectively, and 0.14 V and 0.08 V for $V_{IT}$, respectively. Therefore, $p_s$ will not be perfectly deterministic; instead there will be a small uncertainty in its value, which is represented using the uncertainty band in Fig. 3n. Next, to assess randomness, we utilized the s-bit generator to generate $10^4$ random bits using the same programming and erasing voltage pulses of $V_E = 10$ V and $V_P = -7$ V, respectively, at $\tau_{P/E} = 100$ μs. Supplementary Fig. S7 shows the results of eight of the statistical tests developed by the National Institute of Standards and Technology (NIST) performed on these $10^4$ bits. According to the test protocol, the bitstreams are considered random only if the p-value is greater than 0.01 with the null hypothesis that the sequence is random with 99% confidence level. The NIST test results confirm that the s-bits generated are truly random.

The rough estimate of the energy expenditure for s-bit generation ($E_{s-bit}$) was calculated using Eq. (5).

$$E_{s-bit} = C_G \left( V_P^2 + V_E^2 + V_R^2 + V_{DD}^2 \right) + \langle I_{N1N4} \rangle V_{DD} \tau_{clk} \tag{5}$$

$$\langle I_{N1N4} \rangle = \frac{1}{n} \sum_{i=1}^{n} I_{N1N4,i} \tag{6}$$

$$C_G = \varepsilon_0 \varepsilon_{ox} WL/t_{ox} \tag{7}$$

In Eq. (5), $V_P$, $V_E$, $V_R$, and $V_{DD}$ are the program, erase, read, and supply voltages, respectively. $C_G \approx 10^{-14}$ F is the gate capacitance, $\varepsilon_0 = 8.85 \times 10^{-12} F/m$ is the vacuum permittivity, and $\varepsilon_{ox} = 10$ and $t_{ox} = 50\,nm$ are, respectively, the relative permittivity and thickness of

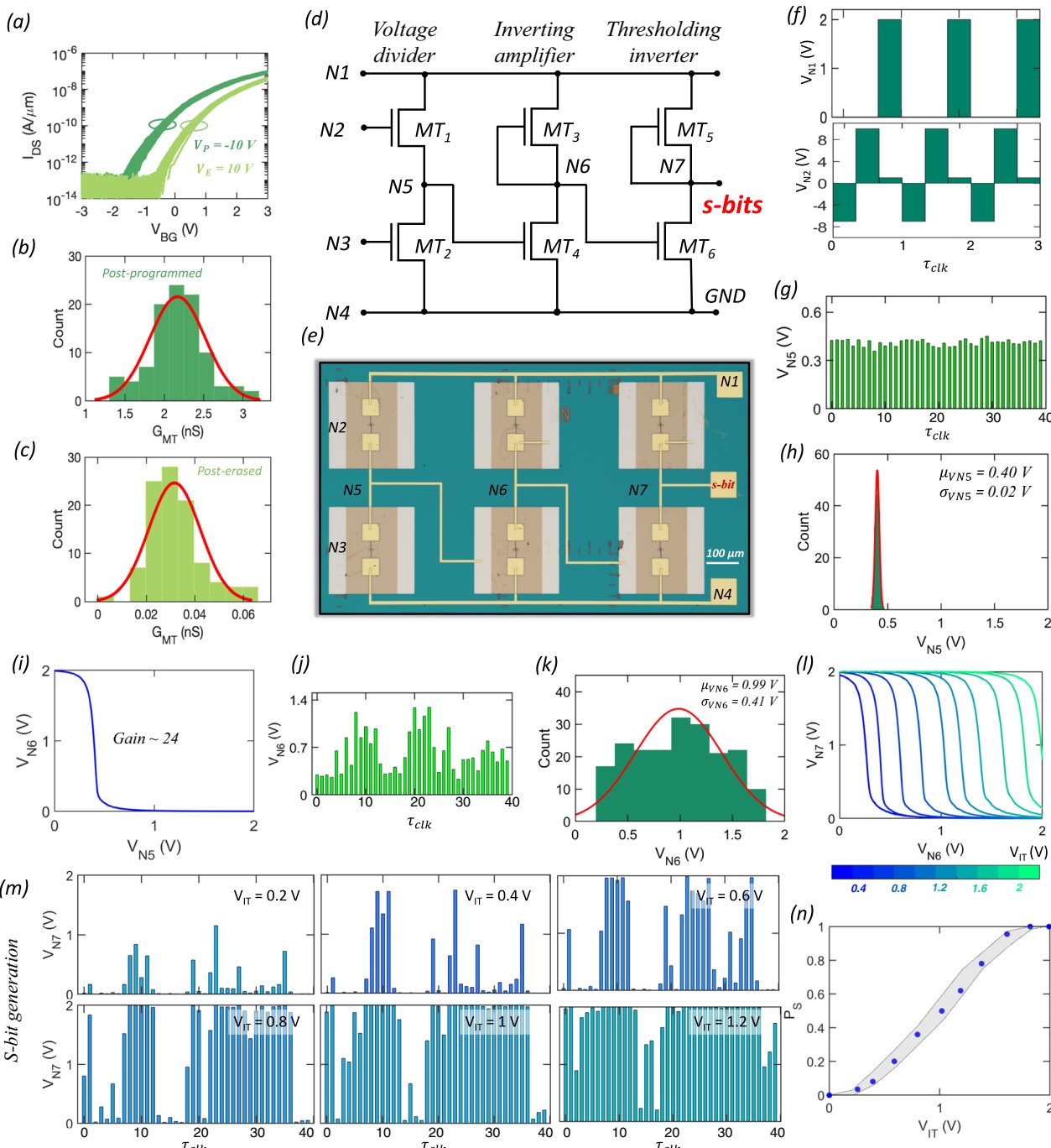

**Fig. 3 | 2D memtransistor-based s-bit generator. a** Transfer characteristics of a representative 2D memtransistor measured after the application of 100 cycles of $V_P = -10\,V$ (dark green) and $V_E = 10\,V$ (light green) pulses, each for $\tau_s = 100\,\mu s$. Distribution of **b** post-programmed and **c** post-erased conductance states ($G_{MT}$) measured using $V_{BG} = 0\,V$. **d** Circuit diagram and **e** corresponding optical image for the proposed s-bit generator consisting of six memtransistors ($MT1$, $MT2$, $MT3$, $MT4$, $MT5$, $MT6$). **f** Voltage waveforms applied to nodes $N1$ and $N2$, i.e., $V_{N1}$ and $V_{N2}$. During each clock cycle ($\tau_{clk}$), $V_{N1}$ toggles between $0\,V$, $0\,V$, and $V_{DD} = 2\,V$ and $V_{N2}$ toggles between $V_P = -7\,V$, $V_E = 10\,V$, and $V_R = 1\,V$. Voltages applied to nodes $N3$ and $N4$, i.e., $V_{N3}$ and $V_{N4}$, are held constant at $1\,V$ and $0\,V$, respectively. **g** Voltage

readout at node $N5$, i.e., $V_{N5}$. **h** Distribution of $V_{N5}$ over $200\ \tau_{clk}$ follows a random Gaussian distribution with a mean ($\mu_{VN5}$) of $0.40\,V$ and a standard deviation ($\sigma_{VN5}$) of $0.02\,V$. **i** Output, $V_{N6}$, of an inverting amplifier constructed using $MT3$ and $MT4$ as a function of the input, $V_{N5}$, with a gain of $-24$. **j** $V_{N6}$ corresponding to $V_{N5}$ shown in **g**. **k** Distribution of $V_{N6}$ follows a random Gaussian distribution with a mean ($\mu_{VN5}$) of $0.99\,V$ and a standard deviation ($\sigma_{VN5}$) of $0.41\,V$. **l** Output, $V_{N7}$, of a thresholding inverter constructed using $MT5$ and $MT6$ as a function of the input, $V_{N6}$, for different inversion threshold, $V_{IT}$. **m** $V_{N7}$ corresponding to $V_{N6}$ shown in **i** for different $V_{IT}$. **n** Probability of obtaining '1' in the bit-stream ($p_s$) as a function of $V_{IT}$.

---

$Al_2O_3$; $W = 5\,\mu m$ and $L = 1\,\mu m$ are, respectively, the channel width and length of the 2D-memtransistor. $\langle I_{N1N4} \rangle$ is the average current flowing through the s-bit generator circuit, i.e., the total current through the voltage divider, inverting amplifier, and threshold inverter during each $\tau_{clk}$. We have used $n = 200$ to calculate the average current per

$\tau_{clk} = 100\,\mu s$ based on the experimental measurements. Since most of the memtransistors operate in their respective subthreshold regimes, the extracted $\langle I_{N1N4} \rangle$ is -1.5 nA as shown in Supplementary Fig. S8. As such, the second term in Eq. (5) accounts for -0.3 pJ, whereas the first term in Eq. (5) accounts for ~2 pJ. This results in $E_{s-bit} \approx 2\,pJ$/clock-cycle,

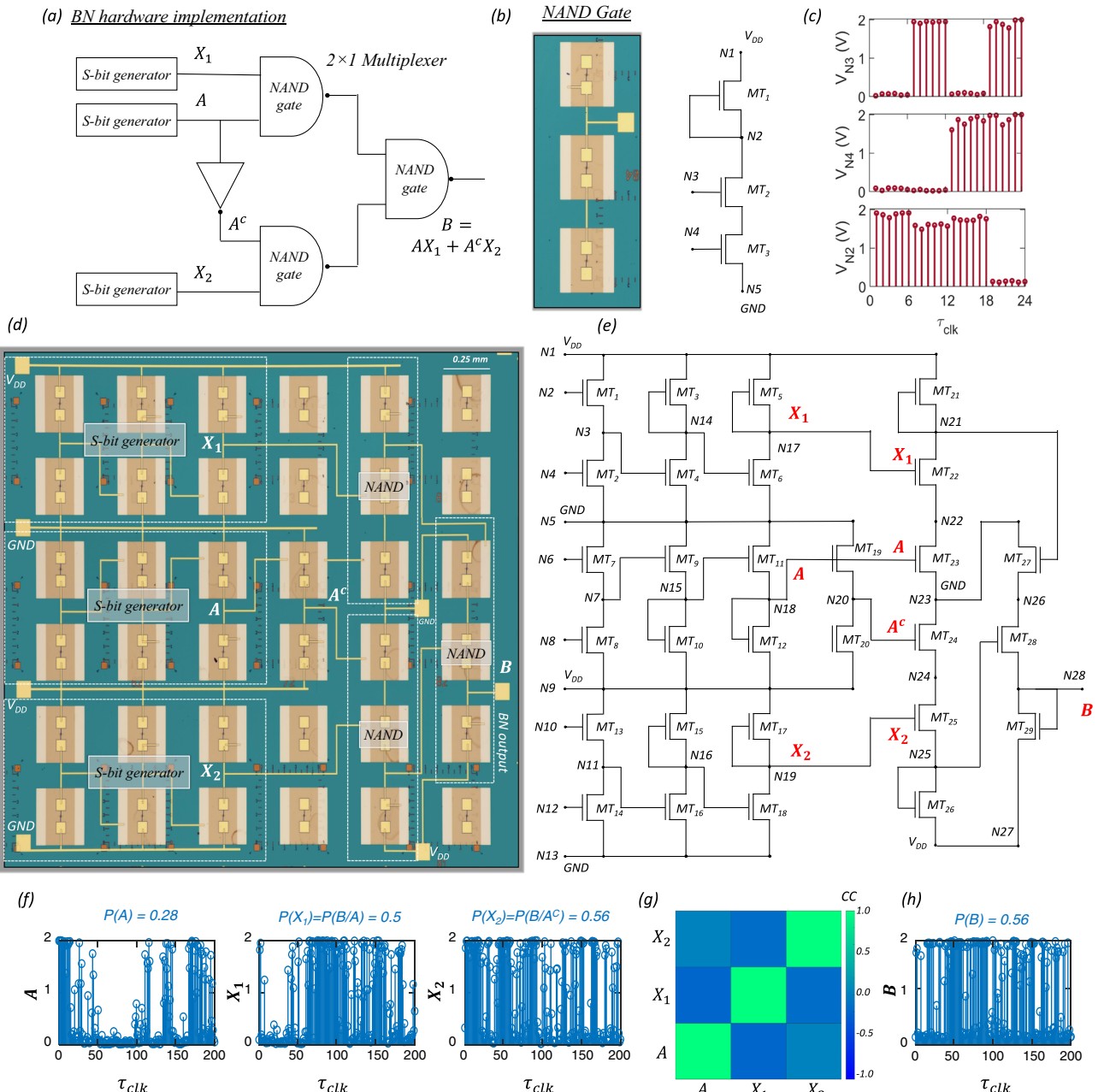

**Fig. 4 | Hardware implementation of BN. a** Circuit schematic for hardware implementation of a BN using three s-bit generators and one 2×1 *MUX*. The *MUX* consists of one inverter and three 2-input *NAND* gates. **b** Optical image and corresponding circuit configuration of a 2-input *NAND* gate comprising 3 memtransistors (*MT*1, *MT*2, and *MT*3) connected in series, with *MT*1 serving as the depletion load. **c** Input waveforms, $V_{N3}$ and $V_{N4}$, which are applied to the local back-gate terminals of *MT*2 and *MT*3 at nodes $N_3$ and $N_4$, respectively, and the corresponding output waveform, $V_{N2}$, which is obtained at node $N_2$. **d** Optical image and **e** corresponding circuit configuration for hardware implementation of a 2-node BN consisting of 3 s-bit generators and a 2×1 *MUX* for a total of 29 memtransistors. The $V_{IT}$ values for the s-bit generators for $X_1$ and $X_2$ can be pre-programmed using the CPT for the nodes A and B. **f** Representative stochastic bit-streams for the random variables $A$, $X_1$, and $X_2$ with $P(A) = 0.28$, $P(X_1) = P(B/A)$ = 0.50, and $P(X_2) = P\left(B/A^C\right) = 0.56$. **g** Correlation coefficient (*CC*) values between $A$, $X_1$, and $X_2$ confirm mutual independence of the s-bit generator modules. **h** Stochastic bit-streams obtained at the output node, $B$. The measured and expected values for $P(B)$ are 0.56 and 0.54, respectively.

which supports our claim of energy efficient s-bit generation. Also note that since each memtransistor has an active device area of ~5 μm², excluding the large contact pads used for probing, the active footprint for the s-bit generator is ~30 μm². Since monolayer 2D materials offer aggressive dimensional scalability, it is possible to reduce the footprint of s-bit generators even further. Nevertheless, the use of only 6 memtransistors is the key towards the realization of area and energy efficient s-bit generator circuits.

**2D memtransistor-based digital circuits and BN implementation**
As described earlier, stochastic multiplexers (*MUX*s) can be used for computing the marginal probability values at any BN node. Figure 4a shows the circuit configuration of a 2×1 *MUX* which consists of one inverter and three 2-input *NAND* gates. Figure 4b shows the optical image and corresponding circuit configuration of a 2-input *NAND* gate comprising 3 memtransistors (*MT*1, *MT*2, and *MT*3) connected in series, with *MT*1 serving as the depletion load. The supply voltage,

$V_{DD}$ = 2 V, is applied to the drain terminal of $MT1$ at node $N_1$, whereas the source terminal of $MT3$, i.e., node $N_5$, is kept grounded. Figure 4c shows the input waveforms, $V_{N3}$ and $V_{N4}$, which are applied to the local back-gate terminals of $MT2$ and $MT3$ at nodes $N_3$ and $N_4$, respectively, and the corresponding output waveform, $V_{N2}$, which is obtained at node $N_2$. Clearly, the circuit operates as a $NAND$ gate.

Figure 4d, e, respectively, show the optical image and corresponding circuit configuration for hardware implementation of a 2-node BN consisting of 3 s-bit generators and a 2 × 1 $MUX$ for a total of 29 memtransistors. The $V_{IT}$ values for the s-bit generators generating $X_1$ and $X_2$ can be pre-programmed corresponding to the CPT for the nodes $A$ and $B$ of the 2-node BN. Figure 4f shows the representative stochastic bit-streams for the random variables $A$, $X_1$, and $X_2$ with $P(A)$ = 0.28, $P(X_1) = P(B/A)$ = 0.50, and $P(X_2) = P\left(B/A^C\right)$ = 0.56. Note that accurate estimation of $P(B)$ requires that the stochastic input variables to the $MUX$, i.e., $A$, $X_1$, and $X_2$, must be mutually independent. Figure 4g shows the correlation coefficient ($CC$) between these three variables. The $CC$ values were found to be close to zero, which confirms mutual independence of the s-bit generator modules. Figure 4h shows the stochastic bit-streams obtained at the output node, $B$. The measured and expected values for $P(B)$ are 0.56 and 0.54, respectively. Supplementary Fig. S9 shows the results for three more sets of measurements. In all instances, we found that our 29 memtransistor module is able to demonstrate a 2-node BN with relatively high accuracy. The rough estimate of the energy expenditure for our hardware BN implementation is miniscule at ~1.2 nJ when 200 $\tau_{clk}$ are used. Certainly, the energy expense can be further reduced by shortening the length of the s-bit streams at the cost of reduced precision. Supplementary Fig. S10 shows the numerical simulation of the error in expected values for $P(B)$ as a function of the bit-length of the s-bit stream for the inputs $P(A)$, $P(B/A)$, and $P(B/A^C)$. The percentage error increases significantly with the reduction in bit-length of the s-bit streams.

While we have experimentally demonstrated that the distribution of the output voltage ($V_{N6}$) from the inverting amplifier follows a Gaussian profile, it is possible that the distribution may deviate from a perfect Gaussian distribution due to many operational reasons. This will definitely lead to computation error. To assess the impact of a skewed distribution on the precision of the BN, we have performed numerical simulations assuming that $V_{N6}$ follows the Pearson random distribution function. Supplementary Fig. S11a shows the distribution of $V_{N6}$ for different values of skewness from −1 to 1 in steps of 0.5. Note that a skewness of −1 or 1 will be a rare occurrence under most practical circumstances. Supplementary Fig. S11b shows the corresponding $p_s$ as a function of $V_{IT}$. As the skewness increases, the deviation of $p_s$ from its expected value also increases. Supplementary Fig. S11c shows the colormap of the percentage error in estimating $P(B)$ using the BN hardware for different skewness in the stochastic input variables $X_1$ and $X_2$ that represent $P(B/A)$ and $P\left(B/A^C\right)$, respectively. As expected, the percentage error increases with increasing skewness. Furthermore, we have experimentally demonstrated that the distribution of the inverting threshold voltage ($V_{IT}$) exhibits a Gaussian distribution after $MT6$ is subjected to 50 program/erase/read cycles with $V_P$ = −7 V, $V_E$ = 10 V, and $\tau_{P/E}$ = 100 μs. This $V_{IT}$ distribution leads to a small uncertainty ($\triangle P$) in probability of output voltages ($V_{N7}$), as shown in Fig. 3n. We have used numerical simulations to assess the impact of uncertainty in obtained probabilities on the precision of the BN, where the probability of the select line, $A$, remains as a constant while the probability of both $X_1$ and $X_2$ are inflicted with $\triangle P$ due to cycle-to-cycle variation in the programmed probability. Supplementary Fig. S12 shows the colormap of the percentage error in estimating $P(B)$ using the BN hardware for uncertainty in the stochastic input variables $X_1$ and $X_2$ that represent $P(B/A)$ = 0.50 and $P\left(B/A^C\right)$ = 0.56, respectively, while $P(A)$ = 0.28 and $\triangle P \approx 0.065$. From this colormap, we can conclude that even if the $V_{IT}$ of the thresholding inverter ($MT6$) is inflicted

with cycle-to-cycle variation from device programming, the inaccuracy of the 2-node Bayesian network ($B = AX_1 + A^C X_2$) is less than 15%. This simulation result shows decent accuracy in hardware implementation of the BN.

Finally, the impact of device-to-device variation on the operation of BN is examined. Supplementary Fig. S13a shows the transfer characteristics of 10 MoS$_2$ memtransistors and Supplementary Fig. S13b shows the transfer characteristics for these 10 devices after one programming/erasing clock cycle ($V_P$ = −7 V, $V_E$ = 10 V, and $\tau_{P/E}$ = 100 μs.). The device-to-device variation translates into error in $\triangle P$ and impacts the accuracy at the output of the BN. Supplementary Fig. S14 shows the colormap of error in $P(B)$ for $P(X_1)$ = 0.5, $P(X_2)$ = 0.56, and $P(A)$ = 0.28. We have used $\triangle P$ =0.046 for both $X_1$ and $X_2$ inferred from Supplementary Fig. S13b. From the error map, it is evident that the variation in the programmed probability inflicted by the device-to-device programming variation of the memtransistors resulted in a maximum error of 8% at the output of the BN.

## Discussion

In conclusion, we have exploited cycle-to-cycle variability in the programmed conductance of 2D memtransistors and transcribed the same into s-bits with reconfigurable probability of obtaining '1' in the bit-stream using a circuit that comprises only 6 memtransistors and spends < 2 pJ per s-bit. We subsequently combined the s-bit generator with a 2D memtransistor-based 2 × 1 $MUX$ to demonstrate hardware implementation of a BN. The BN architecture comprises 29 memtransistors and requires ~ 1.2 nJ of energy for precise computation. Our demonstration of a memtransistor-based standalone in-memory compute fabric shows the potential for emerging 2D materials and devices.

## Methods

### Fabrication of local back-gate islands

To define the back-gate island regions, a commercially-purchased substrate (285 nm SiO$_2$ on p$^{++}$-Si) was spin coated (4000 RPM for 45 s) with bilayer photoresist consisting of Lift-Off-Resist (LOR 5 A) and Series Photoresist (SPR 3012) and baked at 185 °C for 120 s and 95 °C for 60 s, respectively. The bilayer photoresist was then exposed using a Heidelburg Maskless Aligner (MLA 150) to define the island and developed using MF CD26 microposit, followed by a de-ionized (DI) water rinse. The back gate electrode of 20/50 nm TiN/Pt was deposited using reactive sputtering. The photoresist was removed using acetone and Photo Resist Stripper (PRS 3000) and cleaned using 2-propanol (IPA) and DI water. An atomic layer deposition (ALD) process was then implemented to grow 50 nm Al$_2$O$_3$ across the entire substrate, including the island regions. To access the individual Pt back-gate electrodes, etch patterns were defined using the same bilayer photoresist consisting of LOR 5 A and SPR 3012. The bilayer photoresist was then exposed to MLA 150 and developed using MF CD26 microposit. The 50 nm Al$_2$O$_3$ was subsequently dry etched using a BCl$_3$ reactive ion etch (RIE) chemistry at 5 °C for 20 s, which was repeated four times to minimize heating in the substrate. Finally, the photoresist was removed to give access to the individual Pt electrodes.

### Large-area monolayer MoS$_2$ film growth

Monolayer MoS$_2$ was deposited on epi-ready 2″ c-sapphire substrate by metalorganic chemical vapor deposition (MOCVD). An inductively heated graphite susceptor equipped with wafer rotation in a cold-wall horizontal reactor was used to achieve uniform monolayer deposition as previously described[80]. Molybdenum hexacarbonyl (Mo(CO)$_6$) and hydrogen sulfide (H$_2$S) were used as precursors. Mo(CO)$_6$ maintained at 10 °C and 650 Torr in a stainless-steel bubbler was used to deliver $1.1 \times 10^{-3}$ sccm of the metal precursor for the growth, while 400 sccm of H$_2$S was used for the process. MoS$_2$ deposition was carried out at

1000 °C and 50 Torr in $H_2$ ambient, where monolayer growth was achieved in 18 min. The substrate was first heated to 1000 °C in $H_2$ and maintained for 10 min before the growth was initiated. After growth, the substrate was cooled in $H_2S$ to 300 °C to inhibit decomposition of the $MoS_2$ films. More details can be found in our earlier work[45, 48, 81].

### $MoS_2$ film transfer to local back-gate islands

To fabricate the 2D memtransistors, the MOCVD-grown monolayer $MoS_2$ film was transferred from the sapphire growth substrate to the $SiO_2/p^{++}$-Si substrate with local back-gate islands using a PMMA (polymethyl-methacrylate) assisted wet transfer process. First, growth substrate was spin coated with PMMA and left to sit for 24 h to promote PMMA/$MoS_2$ adhesion. The corners of the spin-coated film were scratched using a razor blade and immersed inside 1 M NaOH solution kept at 90 °C. Capillary action caused the NaOH to be drawn into the substrate/film interface, separating the PMMA/$MoS_2$ film from the sapphire substrate. The separated film was rinsed three times inside separate water baths and fished-out using the $SiO_2/p^{++}$-Si substrate with local back-gate islands. The substrate was then baked at 50 °C and 70 °C for 10 min each to remove moisture and promote adhesion. An acetone bath was usd to remove the PMMA supporting layer, with a subsequent IPA bath to remove residue.

### Fabrication of 2D memtransistors

To define the channel regions for the memtransistors, the substrate was spin-coated with PMMA and baked at 180 °C for 90 s. The resist was then patterned using electron beam (e-beam) lithography and developed using a 1:1 mixture of 4-methyl-2-pentanone (MIBK) and 2 propanol (IPA), with a subsequent IPA rinse. The monolayer $MoS_2$ film was then etched using a sulfur hexafluoride ($SF_6$) RIE chemistry at 5 °C for 30 s. Next, the sample was rinsed in acetone and IPA to remove PMMA. To define the source and drain contacts, sample was then spin coated with methyl methacrylate (MMA) followed by PMMA. E-beam lithography was used to pattern the source and drain contacts and 1:1 MIBK/ IPA was again used for development. 40 nm of nickel (Ni) and 30 nm of gold (Au) were deposited using e-beam evaporation. Finally, a lift-off process was performed to remove the excess Ni/Au and resist by immersing the sample in acetone for 30 min followed by IPA for another 30 mins. Each island contains one memtransistor to allow for individual gate control.

### Monolithic integration

To define the connections between respective memtransistors, the substrate was spin coated with MMA and PMMA, followed by e-beam lithography and development using a 1:1 mixture of MIBK/IPA. E-beam evaporation of was used to deposit 60 nm of Ni and 30 nm of Au to form the connections. Finally, the e-beam resist was rinsed away by the same acetone and IPA lift-off process used previously.

### Electrical characterization

Electrical characterization of the fabricated devices was performed using a Lake Shore CRX-VF probe station under atmospheric conditions and with Keysight B1500A parameter analyzer.

## Data availability

The datasets generated during and/or analyzed during the current study are available from the corresponding author on reasonable request.

## Code availability

The codes used for plotting the data are available from the corresponding authors on reasonable request.

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

## Acknowledgements

The work was supported by Army Research Office (ARO) through Contract Number W911NF1920338 and National Science Foundation (NSF) through a CAREER Award under grant no. ECCS-2042154. Authors also acknowledge the materials support from the National Science Foundation (NSF) through the Pennsylvania State University 2D Crystal Consortium–Materials Innovation Platform (2DCCMIP) under NSF cooperative agreement DMR-2039351.

## Author contributions

S.D. conceived the idea and designed the experiments. Y.Z., H.R., and T.F.S. fabricated the memtransistors. Y.Z., H.R., and S.D. performed the measurements, analyzed the data, discussed the results, and agreed on their implications. N.T. grew MOCVD MoS$_2$ under the supervision of J.M.R. All authors contributed to the preparation of the manuscript.

## Competing interests

The authors declare no competing interests.
