## [Peer Review File · Nature Communications]

Title: Hardware Implementation of Bayesian Network based on Two-dimensional MemtransistorsREVIEWER COMMENTS

Reviewer #1 (Remarks to the Author):

This paper proposes a hardware acceleration of BN using a monolithic memtransistor technology based on two-dimensional semiconductors. This technique is a little novel, however, I do not think it can be accepted by this journal. The specific reasons are as follows :

1. The article seems to be a technical improvement in order to complete a task, and the necessary theoretical support is lack.
2. The effect of hardware acceleration is good, however, the improvement is not significant compared to prior art. If the study is specific to existing technological advances, the authors should point out.
3. A large number of information in the graph lacks necessary explanation, and the simple accumulation results make the readability of this paper extremely poor.
4. There are many mistakes in the article that need to be corrected.

Reviewer #2 (Remarks to the Author):

Zheng et al. have experimentally studied the cycle-to-cycle variation in post programmed and post erased conductance states in locally back gated n-type MoS2 transistors. They have demonstrated a stochastic bit generator with a tunable probability of high and low states, made from 6 such transistors, where the stochasticity comes from the cycle-to-cycle variation in one of the transistors and the tunable probability comes from the non-volatile programming of the thresholding inverter formed from two other such transistors. Finally, the authors demonstrate a 2-node Bayesian network made from 29 transistors.

Overall, the idea is of utilizing 2D transistors to form both the stochastic element and the storage element, without the need for peripheral circuitry is potentially interesting. Also, the individual device level characterization presented in the paper is comprehensive. However, in my opinion, the experiments on demonstrating s-bit generation could be supplemented with additional data/analysis to strengthen the study. Further, the proposed MUX based architecture for Bayesian network acceleration needs a clearer analysis of scalability. I have the following points in this regard:

1. On page 5 the last paragraph, the authors mention that "For example, hardware acceleration of BN in Extended Data Fig. 1a can be achieved by using n s-bit generators to obtain the A_i 's, another $N = 2n$ s-bit generators to obtain the CPT, and one $N \times 1$ MUX with n select lines as shown in Extended Data Fig. 1c."

The hardware accelerator for BNs proposed in the literature capture the local statistical dependencies between variables so that the exponentially large space of CPT need not be stored. In the architecture the authors are proposing, the requirement of 2^n s-bit generators to store the CPT would require a prohibitively large hardware resource as n becomes large. Addressing this point appears important for the proposed application in the paper.

2. On page 7, the authors claim that charge trapping and de-trapping processes in the gate dielectric is the source of randomness in the transistors. A bit of detail on how this creates cycle-to-cycle variation in the programmed conductance states would be helpful. Also, please provide references to this for the benefit of the interested reader.

3. The program and erase pulses are applied for a duration of 100 ms. This determines the sampling rate and ultimately the speed of operation of the BN accelerator. What dictates this pulse width and how small can it be made?

4. The random number streams generated from one s-bit generator presented in the paper is around a few hundred (Fig. 3 g-m; Fig. 4 f-h). Can a larger set (104-106) of random numbers be generated from the circuit? This would then enable benchmarking the quality of randomness through tests such as the Statistical Test Suite provided by NIST. Even with the smaller dataset currently presented, a few of the tests in the suite can be performed.

5. The authors have mentioned regarding Fig. 3 b, c that “clearly GMT follows Gaussian distribution”. However, the measured data does not appear like a Gaussian, potentially due to the small data set. It would be useful to discuss why the conductance distribution is expected to be Gaussian.

6. The tunable probability shown in Fig. 3 m, n is achieved through the non-volatile programmability of VIT. How would the VIT be programmed deterministically in a real application, without being affected by the same cycle-to-cycle variation?

Additional comments:

1. The information provided in the first paragraph of the introduction and especially the abstract appears a bit generic and could be focused more on the results of the study.

2. The linear y-axis for Fig. 2c seems to be incorrect (probably missing a $1e-6$ factor).

Reviewer #3 (Remarks to the Author):

In this work, the authors presented the bayesian network using 2D memtransistors. The bayesian network includes probabilistic value, and it is realized by cycle-to-cycle variation (program/erase) as randomness source. I have a couple of comments regarding this manuscript.

1) First of all, I'm not convinced by why this concept should be realized with MoS2 memtransistors. Technically, it can be realized with any 3-terminal memory device which should have cycle-to-cycle variation, and silicon NAND flash will be the most mature technology. Please discuss the cause of MoS2

transistor for this application by comparing with other transistor-type memory devices.

2) In this work, cycle-to-cycle variation is used as randomness source, and I wonder if it would require too many cycles, resulting in a too early computation failure. It looks like a device should undergo program/erase per every clock, which means if the clock speed is 1GHz (lower than very conventional processing unit speed 2-3GHz), there should be 10^9 times of switching. This implies that the proposed device must have pseudo-infinite endurance characteristics like DRAM and unlike flash device. Honestly, it doesn't matter how fast the clock speed is, and it will be a critical issue even with kHz clock. However, the device in this work looks more like flash device and please discuss its feasibility considering endurance characteristics.

3) I wonder what if unlucky case happens for computation like gaussian distribution is not generated despite enough clocks are applied. I think it is very unlikely, however, it should happen sometime if the system is used for a long time and might cause a critical error to entire system.

Reviewer #1 (Remarks to the Author):

This paper proposes a hardware acceleration of BN using a monolithic memtransistor technology based on two-dimensional semiconductors. This technique is a little novel; however, I do not think it can be accepted by this journal. The specific reasons are as follows :

1. The article seems to be a technical improvement in order to complete a task, and the necessary theoretical support is lack.

We would like to thank the reviewer for acknowledging our efforts although we struggle to understand what it means when the reviewer states that our work is “little novel”. Reviewer’s assessment appears more subjective than being based on objective reasoning. The key accomplishment of our work are 1) experimental demonstration of a novel, low-power, and compact s-bit generator circuit based on 2D memtransistor and 2) monolithic integration of 2D memtransistor-based s-bit generator with 2D memtransistor-based logic gates to achieve hardware acceleration of BN, which we feel are significant and worthy of reporting at Nature Communications.

We also wonder what the reviewer means by saying that necessary theoretical support is lacking. The concept of Bayesian network (BN) and stochastic computing (SC) have been extensively studied in computational neuroscience and theoretical computer science. The theoretical basis of BN is widely known and well understood. What is lacking in the community is the experimental demonstration of such computing paradigm based on novel materials, devices, and phenomenon, which can mitigate the energy and area overhead challenges. This is precisely the objective of our work.

2. The effect of hardware acceleration is good; however, the improvement is not significant compared to prior art. If the study is specific to existing technological advances, the authors should point out.

We would like to thank the reviewer for his appreciation of the hardware acceleration of BN. However, we respectfully disagree with their statement that the improvements are not significant compared to prior art. For example, probabilistic CMOS based implementation of BN consumes

~ 150 pJ energy and requires multiple hardware elements including amplifiers, inverters, flip-flops etc. to generate the s-bits, which increases the transistor count to more than 100 making it energy area inefficient [1]. Similarly, BN acceleration using field programmable gate arrays (FPGA)[2-4] also require large number of transistors that consume significant amount of energy to generate random bits owing to the absence of inherent stochasticity in silicon devices. Stochastic switching in memristors offer an excellent mechanism to generate random bits [5]. While memristors can be aggressively scaled [6], experimentally reported true random number generator (TRNGs) based on memristors [7] have large footprint ~ 25 μm^2 excluding the CMOS peripherals that included one comparator, one AND gate, and two 4-bit counters (>100 CMOS transistors) and the energy consumption was found to be 15 pJ/bit. Finally, stochastic magnetic tunnel junctions (MTJs) can generate s-bits at a frugal energy expenditure of 2 fJ/bit every 1-100 ms. While the footprint of MTJs can be rather small, generation of s-bits require integration with CMOS peripherals such as one resistor, one transistor, and one comparator, which increases the design complexity and negates the energy and area benefits. Furthermore, two-terminal memristor and spintronics devices are incapable of performing logic operations necessitating hybrid design involving CMOS peripheral for full demonstration of BN. In contrast, our s-bit generator comprises of only 6 2D memtransistors and consumes only 2 pJ/bit and we also eliminate the need for hybrid design since three-terminal memtransistors can be utilized as logic elements.

3. A large number of information in the graph lacks necessary explanation, and the simple accumulation results make the readability of this paper extremely poor.

We apologize for lack of explanation on figures. We have added explanation and improved readability of the manuscript.

4. There are many mistakes in the article that need to be corrected.

The reviewer's concern is noted. We have carefully read the manuscript and made necessary corrections.

Reviewer #2 (Remarks to the Author):

Zheng et al. have experimentally studied the cycle-to-cycle variation in post programmed and post erased conductance states in locally back gated n-type MoS₂ transistors. They have demonstrated a stochastic bit generator with a tunable probability of high and low states, made from 6 such transistors, where the stochasticity comes from the cycle-to-cycle variation in one of the transistors and the tunable probability comes from the non-volatile programming of the thresholding inverter formed from two other such transistors. Finally, the authors demonstrate a 2-node Bayesian network made from 29 transistors. Overall, the idea is of utilizing 2D transistors to form both the stochastic element and the storage element, without the need for peripheral circuitry is potentially interesting. Also, the individual device level characterization presented in the paper is comprehensive. However, in my opinion, the experiments on demonstrating s-bit generation could be supplemented with additional data/analysis to strengthen the study. Further, the proposed MUX based architecture for Bayesian network acceleration needs a clearer analysis of scalability. I have the following points in this regard:

We would like to thank the reviewer for his thorough reading and acknowledgment of our work. We are happy to provide further analysis and evidence on s-bit generation and MUX-based architecture for the Bayesian network (BN).

1. On page 5 the last paragraph, the authors mention that “For example, hardware acceleration of BN in Extended Data Fig. 1a can be achieved by using n s-bit generators to obtain the A_i 's, another $N = 2n$ s-bit generators to obtain the CPT, and one $N \times 1$ MUX with n select lines as shown in Extended Data Fig. 1c.” The hardware accelerator for BNs proposed in the literature capture the local statistical dependencies between variables so that the exponentially large space of CPT need not be stored. In the architecture the authors are proposing, the requirement of 2^n s-bit generators to store the CPT would require a prohibitively large hardware resource as n becomes large. Addressing this point appears important for the proposed application in the paper.

We completely agree with the reviewer that for most realistic scenarios, where Bayesian networks (BN) can be used, it is highly unlikely that a child node will have a large number of parent nodes (n) or vice versa. Since BN represents a set of variables and their conditional dependencies via a directed acyclic graph, finding an optimal graphical representation is critical in minimizing the hardware resources necessary for its acceleration. Local statistical dependencies between the variables are often used to simplify the graphical representation for BN [8]. For example, graphs in which the edges are oriented according to a causal theory are generally more efficient [9]. Therefore, the number of s-bit generators that will be required to store the CPT will remain small for all practical purposes. We have revised the **Extended Data Fig. 1** and the associated discussion to avoid the confusion for the readers.

2. On page 7, the authors claim that charge trapping and de-trapping processes in the gate dielectric is the source of randomness in the transistors. A bit of detail on how this creates cycle-to-cycle variation in the programmed conductance states would be helpful. Also, please provide references to this for the benefit of the interested reader.

Reviewer's suggestion is noted. We have included the following discussions on the phenomenon of charge trapping and de-trapping at and near the dielectric/2D interface and the origin of cycle-to-cycle variation in the programmed conductance states in the revised manuscript along with relevant references.

The shift in the transfer characteristics of post-programmed and post-erased 2D memtransistors can be explained using the phenomenon of charge trapping and de-trapping at and near the 2D/dielectric interface. Note that trap states can originate from defects/imperfections in the dielectric and/or adsorbed species at the 2D/dielectric interface as reported in various earlier studies [10-12]. These states can also be engineered at desired energetic locations by introducing intentional defects in the 2D channel material [13, 14]. Carrier occupancy in these trap states follow Fermi-Dirac distribution. As illustrated using the energy band diagrams in **Fig. R1**, at equilibrium, i.e. in the absence of any gate bias, the trap states with energy levels above the Fermi energy (E_F) are empty, whereas the ones below E_F are filled. When the memtransistor is subjected to a negative "Write" (V_P) voltage pulse, electrons are released (de-trapped) from these trap states

Figure R1. Energy band diagram explaining the charge trapping phenomena while programming the 2D memtransistor.

leaving them positively charged. This leads to screen of the back-gate bias, which is reflected as shift in the threshold voltage (ΔV_{TH}) following Eq. 1.

$$\Delta V_{TH} = -\frac{Q_T}{C_{OX}}; C_{OX} = \frac{\epsilon_{OX}}{t_{OX}} \quad [1]$$

Where, Q_T is the magnitude of effective positive charge at or near the 2D/dielectric interface, and C_{OX} is the capacitance, ϵ_{OX} is the dielectric constant ($10\epsilon_0$, $\epsilon_0 = 8.85 \times 10^{-12}$ F/m²), and $t_{OX} = 50$ nm is the thickness of the back gate oxide, respectively. Similarly, when the memtransistor is subjected to a positive “Erase” (V_E) voltage pulse, electrons are captured back (trapped) into the trap states restoring the threshold voltage. Note that the number of electrons getting trapped/de-trapped can be controlled by both the magnitude and duration of V_P and V_E , which allow us to have an analog control of the ΔV_{TH} or the conductance state of the memtransistor.

The cycle-to cycle variation in program/erase processes is a direct consequence of the stochastic nature of charge trapping and de-trapping observed in most semiconductor/dielectric interfaces [15-17]. In the simple two-state model, a trap state can be electrically neutral or charged and can transition between the two states even under equilibrium condition with transition times exponentially distributed. In other words, the state transition dynamics for traps follows classic Markovian process [18, 19]. In ultra-scaled metal oxide semiconductor field effect transistors (MOSFETs) such stochastic state transitions lead to random telegraph noise (RTN). Often metastable states are also involved in the trapping/detrapping processes making the transition

dynamic more complex, rich, and at the same time introducing additional source of randomness [20]. While RTN is not observed in our relatively large area memtransistors, the stochasticity of trapping/detrapping processes manifest during the program/erase operations leading to the cycle-to-cycle variation in ΔV_{TH} . Note that a detailed discussion on the origin of stochasticity is beyond the scope of this article and interested readers can find more information in the literature.

3. The program and erase pulses are applied for a duration of 100 ms. This determines the sampling rate and ultimately the speed of operation of the BN accelerator. What dictates this pulse width and how small can it be made?

The reviewer has raised an excellent point. The minimum program/erase pulse width is determined by the trapping/detrapping time constants, which can be as short as several hundreds of

Figure R2. Post-programmed and Post-erased transfer characteristics of a 2D memtransistor subjected to negative “Write” (V_P) and positive “Erase” (V_E) voltage pulses of different amplitudes ranging from 8 V to 15 V applied to the local back-gate electrode, each for a duration of a) $\tau_{P/E} = 100 \mu\text{s}$, b) $\tau_{P/E} = 10 \mu\text{s}$, c) $\tau_{P/E} = 1 \mu\text{s}$, and d) $\tau_{P/E} = 100 \text{ ns}$. Extracted shift in the threshold voltage (ΔV_{TH}) as a function of $V_{P/E}$ for e) $\tau_{P/E} = 100 \mu\text{s}$ and f) $\tau_{P/E} = 100 \text{ ns}$.

picoseconds. Therefore, it is possible to reduce the pulse width. We have now performed additional experiments on the programmability of 2D memtransistors with different pulse widths, ranging from 100 μs down to 100 ns. **Fig. R2a-d** show the post-programmed and post-erased transfer characteristics of a 2D memtransistor subjected to negative “Write” (V_P) and positive “Erase” (V_E) voltage pulses of different amplitudes ranging from 8 V to 15 V applied to the local back-gate electrode, each for a duration of $\tau_{P/E} = 100 \mu\text{s}$, 10 μs , 1 μs , and 100 ns, respectively. Clearly, the charge trapping and detrapping process can occur as fast as 100 ns. Therefore, our BN accelerator can be made to operate fast. Also note that the fastest pulse timing was limited by our measurement tools. **Fig. R2e-f**, respectively, show the extracted shift in the threshold voltage (ΔV_{TH}) as a function of τ_P for $V_P = -8$ V and -15 V and τ_E for $V_E = 8$ V and 15 V, respectively. From these results, we can conclude that, for any given pulse magnitude $V_{P/E}$, ΔV_{TH} becomes smaller as $\tau_{P/E}$ becomes shorter. To retain similar ΔV_{TH} for smaller $\tau_{P/E}$ larger $V_{P/E}$ is required, which will increase the energy expenditure. Therefore, one needs to strike a balance between fast programmability and energy consumption based on the application.

4. The random number streams generated from one s-bit generator presented in the paper is around a few hundred (Fig. 3 g-m; Fig. 4 f-h). Can a larger set (10^4 - 10^6) of random numbers be generated from the circuit? This would then enable benchmarking the quality of randomness through tests such as the Statistical Test Suite provided by NIST. Even with the smaller dataset currently presented, a few of the tests in the suite can be performed.

The reviewer has an excellent point here. The s-bit generator can be used to generate larger set of random bits. We have utilized the s-bit generator to generate 10^4 random bits using the same programming and erase voltage pulses of $V_E = 10$ V and $V_P = -7$ V, respectively, and $\tau_{P/E} = 100 \mu\text{s}$. **Table. R1** shows the results of eight of the statistical tests developed by the National Institute of Standards and Technology performed on 10^4 bits obtained from our s-bit generator. According to the test protocol, the bits streams are considered random only if the p-value is greater than 0.01 with the null hypothesis that the sequence is random with 99% confidence level. It is evident from the results that the s-bits generated are truly random.

Table R1		
NIST Test	p-value	Result
Frequency Monobit Test	0.0357	Pass
Frequency Test within a Block	0.3933	Pass
Runs Test	0.1433	Pass
Longest Run of Ones	0.0935	Pass
Binary Matrix Rank Test	0.3230	Pass
Discrete Fourier Transform Test	0.0669	Pass
Serial Test	0.1561	Pass
Cumulative Sums (Cusum) Test	0.3673	Pass

5. The authors have mentioned regarding Fig. 3b-c that “clearly GMT follows Gaussian distribution”. However, the measured data does not appear like a Gaussian, potentially due to the small data set. It would be useful to discuss why the conductance distribution is expected to be Gaussian.

The reviewer has raised two valid concerns. It is indeed true that a better Gaussian fitting can be obtained when the data set is large. As mentioned in the manuscript, to translate the stochastic conductance fluctuation in post-programmed 2D memtransistors into s-bits, we deploy a circuit consisting of six memtransistors ($MT1$, $MT2$, $MT3$, $MT4$, $MT5$, and $MT6$) as shown using the circuit diagram in **Fig. R3a**. Note that during each clock cycle (τ_{clk}), V_{N1} switches between 0 V, 0 V, and 2 V and V_{N2} switches between $V_P = -7$ V, $V_E = 10$ V, and $V_R = 1$ V. Voltages applied to nodes, $N3$, and $N4$, i.e., V_{N3} , and V_{N4} are held constant at 1V and 0 V, respectively. This allows programming and reset of $MT1$ during each τ_{clk} . The distribution of voltage readout at node, $N5$, i.e., V_{N5} is shown in **Fig. R3b**. Note that the series connection of memtransistors, $MT1$ and $MT2$ represents a voltage divider circuit, and hence V_{N5} is determined by their respective conductance values, i.e., G_{MT1} and G_{MT2} . Since G_{MT1} fluctuates from cycle-to-cycle owing to programming and reset voltages applied to its local back-gate terminal, i.e., $N2$, so does V_{N5} . In other words, the voltage divider translates conductance fluctuation into voltage fluctuation. Here, we have used 10^4

programming/erase/read cycles to obtain larger data set for cycle-to-cycle variation in V_{N5} . Clearly, the distribution fits well with Gaussian profile.

While, it is difficult to explain why the conductance distribution is expected to be Gaussian, the experimental results tend to fit well with such distribution. Similar observation is made in the literature on other material systems that involve charge trapping/detrapping. For example, the random telegraph noise (RTN) in scaled resistive random access memory (RRAM) devices originating from activation/deactivation of the electron traps in the filament also leads to conductance fluctuation that follows Gaussian distribution [21]. RTN in small semiconductor devices also lead to conductance fluctuation that follows Gaussian distribution [22]. However, there are evidence of skewed non-Gaussian distributions originating from RTN [23]. We would invest in detailed theoretical modeling of the phenomenon in the future to explain the origin of Gaussian distribution.

Figure R3. a) The s -bit generator circuit based on 2D memtransistors. b) Distribution of V_{N5} when 10^4 programming/erase/read cycles are used. Clearly, the distribution fits well with Gaussian profile.

6. The tunable probability shown in Fig. 3 m, n is achieved through the non-volatile programmability of VIT. How would the VIT be programmed deterministically in a real application, without being affected by the same cycle-to-cycle variation?

The reviewer has raised an excellent point. We agree that the cycle-to-cycle variation in the programming of 2D memtransistors will lead to fluctuations in the threshold voltage (V_{TH}) of MT_6

and hence in V_{IT} of the thresholding inverter and p_s for the s-bit stream. **Fig. R4a-b**, respectively, show the distribution of V_{TH} and V_{IT} when MT_6 is subjected to 50 program/erase/read cycles with $V_P = -7V$, $V_E = 10 V$ and $\tau_{P/E} = 100 \mu s$. The mean and standard deviation values were found to be $-0.04 V$ and $0.08 V$ for V_{TH} , $0.14 V$ and $0.08 V$ for V_{IT} . This leads to variation in the p_s as shown as a band of uncertainty around the mean value of p_s in **Fig. R4c**. Therefore, we cannot claim that p_s will be perfectly deterministic, instead there will be small uncertainty in its value. We have revised **Fig. 3n** and included this uncertainty band and commented on the same in the revised manuscript.

Figure R4. a) Distribution of cycle-to-cycle programming variation in the V_{TH} of MT_6 and b) the corresponding variation in the V_{IT} of the thresholding inverter constructed using MT_5 and MT_6 when MT_6 is subjected to 50 program/erase/read cycles with $V_P = -7V$, $V_E = 10 V$ and $\tau_{P/E} = 100 \mu s$. c) The corresponding variation in the p_s shown as a band of uncertainty around its mean value.

Additional comments:

1. The information provided in the first paragraph of the introduction and especially the abstract appears a bit generic and could be focused more on the results of the study.

We agree with the reviewer's suggestions. We have revised the abstract and introduction accordingly.

2. The linear y-axis for Fig. 2c seems to be incorrect (probably missing a $1e-6$ factor).

We are sorry for the typo. We have corrected the labels for Fig. 2c.

Reviewer #3 (Remarks to the Author):

In this work, the authors presented the Bayesian network using 2D memtransistors. The Bayesian network includes probabilistic value, and it is realized by cycle-to-cycle variation (program/erase) as randomness source. I have a couple of comments regarding this manuscript.

1) First of all, I'm not convinced by why this concept should be realized with MoS₂ memtransistors. Technically, it can be realized with any 3-terminal memory device which should have cycle-to-cycle variation, and silicon NAND flash will be the most mature technology. Please discuss the cause of MoS₂ transistor for this application by comparing with other transistor-type memory devices.

The reviewer is absolutely correct that the s-bit generators, in principle, can be realized using any three-terminal memory devices that show cycle-to-cycle programming variation. In fact, commercial silicon NAND Flash memory devices have been explored as high quality true random number generators (TRNG) [24, 25]. However, instead of cycle-to-cycle programming variation, NAND based TRNGs exploit thermal noise, random telegraph noise (RTN) and device-to-device variation in the threshold voltage. Furthermore, a digital interface involving silicon CMOS-based peripherals is also necessary for converting the analog fluctuations into s-bits with reconfigurable probability of obtaining '1's or '0's in a given bit stream. Therefore, despite their maturity, the NAND flash and CMOS technology based on silicon do not offer a monolithically solution today. This is the so-called von Neumann bottleneck, which highlights the importance of 'in-memory compute' for energy and hardware efficient acceleration of emerging computing paradigms including stochastic computing, brain-inspired neuromorphic computing, etc. And our demonstration of compact s-bit generator based on only six 2D memtransistors reinforces this claim.

The choice of large-area grown monolayer MoS₂ is motivated by the fact that atomically thin 2D materials are being seriously considered by the semiconductor industry (for example, companies like TSMC and Intel) for advanced technology nodes [26]. It is widely accepted that scaling silicon thickness beyond ~ 3-4 nm is challenging. Yet, the gate electrostatics demand aggressive reduction in the channel thickness to preserve the desired device performance for sub-10 nm technology

nodes [27]. The ultimate channel thickness that one can envision for a field-effect transistor (FET) would be in the sub-1 nm range, which is not readily accessible for any three-dimensional (3D) semiconducting crystal due to increased scattering of charge carriers at the channel-to-dielectric interfaces that results in severe mobility degradation [28]. This opens up opportunities for semiconducting two-dimensional (2D) materials, which are naturally thin with monolayers having sub-1 nm (~ 0.6 nm) body thicknesses in case of transition metal dichalcogenides (TMDs) [29-35]. Moreover, the absence of dangling bonds offers the potential to achieve better channel-to-dielectric interfaces. Note that some of the early criticism of 2D FETs have been successfully addressed through the realization of low contact resistance (R_C) [36], high ON current [37], integration of ultra-thin and high-k gate dielectric [38], and wafer scale growth using metal organic chemical vapor deposition (MOCVD) technique [39]. Recent experimental breakthroughs in the development of high-performance 2D field effect transistors (FETs) [36-39], neurosynaptic devices [40-44] and very large scale integrated (VLSI) circuits [14, 45-47] also establishes the fact that 2D monolayers have tremendous potential. Similarly, theoretical calculations and quantum mechanical simulation have found that the 2D vdW FETs can outperform CMOS HP (high performance) in both energy and delay [48-51].

Based on the above discussion, our demonstration of a standalone hardware platform exploiting 2D memtransistors not only shows the promise of ‘in-memory compute’ for energy and hardware efficient acceleration of novel computing paradigms but also highlights the long-term benefits to the semiconductor eco-system and road to resolve grad challenges that exists with the aging silicon technology. However, we also acknowledge that a significant amount of work will be necessary before 2D materials are introduced in commercial products and there are scopes for improvement and optimization of the 2D memtransistor technology. For example, large-area growth [26] and large-area transfer [35] of 2D materials must be perfected to minimize growth defects and damages caused due to transfer to ensure high yield during device fabrication. Finally, the programmability of scaled 2D memtransistors must be investigated to ensure that these devices can meet the requirements set forth by the International Roadmap for Devices and Systems (IRDS 2028).

2) In this work, cycle-to-cycle variation is used as randomness source, and I wonder if it would require too many cycles, resulting in a too early computation failure. It looks like a

device should undergo program/erase per every clock, which means if the clock speed is 1GHz (lower than very conventional processing unit speed 2-3GHz), there should be 10^9 times of switching. This implies that the proposed device must have pseudo-infinite endurance characteristics like DRAM and unlike flash device. Honestly, it doesn't matter how fast the clock speed is, and it will be a critical issue even with kHz clock. However, the device in this work looks more like flash device and please discuss its feasibility considering endurance characteristics.

The reviewer's concern of endurance of the device is completely valid and we do agree that it is unlikely that the 2D memtransistor will have pseudo-infinite endurance characteristics like the DRAM. We have now conducted endurance experiments on our MoS₂ memtransistor with the gate voltage cycle of $V_P = -7$ V and $V_E = 10$ V with $\tau_{P/E} = 100$ ns up to 10^9 cycles. **Fig. R5** shows the post-program and post-erase conductance measured at $V_{BG} = 0$ V for up to 10^9 endurance cycles. There is no significant change in the two states. While, we will continue to test the endurance of our memtransistor for higher number of cycles in our future studies, we think for the applications that we sought for, i.e., edge computing, our technology will still be extremely useful. Edge applications significantly reduce endurance requirements to achieve energy and resource efficiency. For example, in weather forecasting, the BN will be used every minute rather than every microseconds, similarly, in medical diagnostics, BN will be used only several thousand times a day to assess patients.

Figure R5. Programming endurance for 2D memtransistor for 10^9 cycles.

3) I wonder what if unlucky case happens for computation like gaussian distribution is not generated despite enough clocks are applied. I think it is very unlikely, however, it should happen sometime if the system is used for a long time and might cause a critical error to entire system.

The reviewer has raised a valid concern. It is indeed possible that the distribution of the output voltage (V_{N5}) of the divider circuit constructed using $MT1$ and $MT2$ may not necessarily follow a perfect Gaussian distribution even after the clock is applied for enough cycles. This will definitely lead to computation error. To assess the impact of skewed distribution on the precision of the BN, we have performed simulations using MATLAB assuming that V_{N6} , i.e., the output of the inverting amplifier of the s -bit generator circuit follows the Pearson random distribution function. **Fig. R6a** shows the distribution of V_{N6} for different values of skewness from -1 to 1 in steps of 0.5. **Fig. R6b** shows the corresponding p_s as a function of V_{IT} . As the skewness increases, the deviation of p_s from its expected value also increases. **Fig. R6c** shows the colormap of the percentage error in estimating $P(B)$ using the BN accelerator for different skewness in the stochastic input variables X_1 and X_2 that represent $P(B/A)$ and $P(B/A^C)$, respectively. As expected, the percentage error increases with increasing skewness. We have added the above discussion in the **Extended Data Fig. 9**.

Figure R6. a) Simulation results showing V_{N6} , i.e., the output of the inverting amplifier of the s -bit generator circuit drawn from Pearson random distribution function with different skewness. b) Corresponding p_s as a function of V_{IT} . As the skewness increases, the deviation of p_s from its expected value also increases. c) Colormap of the percentage error in estimating $P(B)$ using the BN accelerator for different skewness in the stochastic input variables X_1 and X_2 that represent $P(B/A)$ and $P(B/A^C)$, respectively.

Reference:

- [1] Z. Weijia, G. W. Ling, and Y. K. Seng, "PCMOS-based Hardware Implementation of Bayesian Network," in *2007 IEEE Conference on Electron Devices and Solid-State Circuits*, 2007, pp. 337-340.
- [2] R. Cai, A. Ren, N. Liu, C. Ding, L. Wang, X. Qian, *et al.*, "VIBNN: Hardware Acceleration of Bayesian Neural Networks," presented at the Proceedings of the Twenty-Third International Conference on Architectural Support for Programming Languages and Operating Systems, Williamsburg, VA, USA, 2018.
- [3] Z. Kulesza and W. Tylman, "Implementation Of Bayesian Network In FPGA Circuit," in *Proceedings of the International Conference Mixed Design of Integrated Circuits and System, 2006. MIXDES 2006.*, 2006, pp. 711-715.
- [4] S. Zermani, C. Dezan, H. Chenini, J. Diguët, and R. Euler, "FPGA implementation of Bayesian network inference for an embedded diagnosis," in *2015 IEEE Conference on Prognostics and Health Management (PHM)*, 2015, pp. 1-10.
- [5] W. Sun, B. Gao, M. Chi, Q. Xia, J. J. Yang, H. Qian, *et al.*, "Understanding memristive switching via in situ characterization and device modeling," *Nature communications*, vol. 10, pp. 1-13, 2019.
- [6] S. Pi, C. Li, H. Jiang, W. Xia, H. Xin, J. J. Yang, *et al.*, "Memristor crossbar arrays with 6-nm half-pitch and 2-nm critical dimension," *Nature Nanotechnology*, vol. 14, pp. 35-39, 2019/01/01 2019.
- [7] H. Jiang, D. Belkin, S. E. Savel'ev, S. Lin, Z. Wang, Y. Li, *et al.*, "A novel true random number generator based on a stochastic diffusive memristor," *Nature Communications*, vol. 8, p. 882, 2017/10/12 2017.
- [8] R. G. Almond, J. Mulder, L. A. Hemat, and D. Yan, "Bayesian network models for local dependence among observable outcome variables," *Journal of Educational and Behavioral Statistics*, vol. 34, pp. 491-521, 2009.
- [9] C. Glymour, "Learning, prediction and causal Bayes nets," *Trends in cognitive sciences*, vol. 7, pp. 43-48, 2003.
- [10] A. J. Arnold, A. Razavieh, J. R. Nasr, D. S. Schulman, C. M. Eichfeld, and S. Das, "Mimicking Neurotransmitter Release in Chemical Synapses via Hysteresis Engineering in MoS₂ Transistors," *ACS nano*, vol. 11, pp. 3110-3118, 2017.
- [11] Y. Y. Illarionov, G. Rzepa, M. Waltl, T. Knobloch, A. Grill, M. M. Furchi, *et al.*, "The role of charge trapping in MoS₂/SiO₂ and MoS₂/hBN field-effect transistors," *2D Materials*, vol. 3, 2016.
- [12] Y. Y. Illarionov, T. Knobloch, M. Waltl, G. Rzepa, A. Pospischil, D. K. Polyushkin, *et al.*, "Energetic mapping of oxide traps in MoS₂ field-effect transistors," *2D Materials*, vol. 4, p. 025108, 2017.
- [13] J. Jiang, C. Ling, T. Xu, W. Wang, X. Niu, A. Zafar, *et al.*, "Defect engineering for modulating the trap states in 2D photoconductors," *Advanced Materials*, vol. 30, p. 1804332, 2018.
- [14] S. Das, A. Sebastian, E. Pop, C. J. McClellan, A. D. Franklin, T. Grasser, *et al.*, "Transistors based on two-dimensional materials for future integrated circuits," *Nature Electronics*, vol. 4, pp. 786-799, 2021/11/01 2021.
- [15] S. Bhagdikar and S. Mahapatra, "A Stochastic Hole Trapping-Detrapping Framework for NBTI, TDDS and RTN," in *2019 International Conference on Simulation of Semiconductor Processes and Devices (SISPAD)*, 2019, pp. 1-4.

- [16] T. Grasser, "Stochastic charge trapping in oxides: From random telegraph noise to bias temperature instabilities," *Microelectronics Reliability*, vol. 52, pp. 39-70, 2012/01/01/2012.
- [17] M. Waltl, "Characterization and Modeling of Single Charge Trapping in MOS Transistors," in *2019 IEEE International Integrated Reliability Workshop (IIRW)*, 2019, pp. 1-9.
- [18] M. Kirton and M. Uren, "Noise in solid-state microstructures: A new perspective on individual defects, interface states and low-frequency (1/f) noise," *Advances in Physics*, vol. 38, pp. 367-468, 1989.
- [19] O. Ibe, *Markov processes for stochastic modeling*: Newnes, 2013.
- [20] M. Uren, M. Kirton, and S. Collins, "Anomalous telegraph noise in small-area silicon metal-oxide-semiconductor field-effect transistors," *Physical Review B*, vol. 37, p. 8346, 1988.
- [21] D. Veksler, G. Bersuker, L. Vandelli, A. Padovani, L. Larcher, A. Muraviev, *et al.*, "Random telegraph noise (RTN) in scaled RRAM devices," in *2013 IEEE International Reliability Physics Symposium (IRPS)*, 2013, pp. MY. 10.1-MY. 10.4.
- [22] Y. Yuzhelevski, M. Yuzhelevski, and G. Jung, "Random telegraph noise analysis in time domain," *Review of Scientific Instruments*, vol. 71, pp. 1681-1688, 2000.
- [23] C. Y.-P. Chao, H. Tu, T. Wu, K.-Y. Chou, S.-F. Yeh, and F.-L. Hsueh, "CMOS image sensor random telegraph noise time constant extraction from correlated to uncorrelated double sampling," *IEEE Journal of the Electron Devices Society*, vol. 5, pp. 79-89, 2016.
- [24] B. Ray and A. Milenković, "True random number generation using read noise of flash memory cells," *IEEE transactions on electron devices*, vol. 65, pp. 963-969, 2018.
- [25] Y. Wang, W.-k. Yu, S. Wu, G. Malysa, G. E. Suh, and E. C. Kan, "Flash memory for ubiquitous hardware security functions: True random number generation and device fingerprints," in *2012 IEEE Symposium on Security and Privacy*, 2012, pp. 33-47.
- [26] M.-Y. Li, S.-K. Su, H.-S. P. Wong, and L.-J. Li, "How 2D semiconductors could extend Moore's law," ed: Nature Publishing Group, 2019.
- [27] A. P. Jacob, R. Xie, M. G. Sung, L. Liebmann, R. T. Lee, and B. Taylor, "Scaling challenges for advanced CMOS devices," *International Journal of High Speed Electronics and Systems*, vol. 26, p. 1740001, 2017.
- [28] K. Uchida, H. Watanabe, A. Kinoshita, J. Koga, T. Numata, and S. Takagi, "Experimental study on carrier transport mechanism in ultrathin-body SOI nand p-MOSFETs with SOI thickness less than 5 nm," in *Digest. International Electron Devices Meeting*, 2002, pp. 47-50.
- [29] S. Manzeli, D. Ovchinnikov, D. Pasquier, O. V. Yazyev, and A. Kis, "2D transition metal dichalcogenides," *Nature Reviews Materials*, vol. 2, p. 17033, 2017.
- [30] D. Akinwande, C. Huyghebaert, C.-H. Wang, M. I. Serna, S. Goossens, L.-J. Li, *et al.*, "Graphene and two-dimensional materials for silicon technology," *Nature*, vol. 573, pp. 507-518, 2019.
- [31] M. Chhowalla, D. Jena, and H. Zhang, "Two-dimensional semiconductors for transistors," *Nature Reviews Materials*, vol. 1, pp. 1-15, 2016.
- [32] F. Schwier, J. Pezoldt, and R. Granzner, "Two-dimensional materials and their prospects in transistor electronics," *Nanoscale*, vol. 7, pp. 8261-8283, 2015.

- [33] C. Liu, H. Chen, S. Wang, Q. Liu, Y.-G. Jiang, D. W. Zhang, *et al.*, "Two-dimensional materials for next-generation computing technologies," *Nature Nanotechnology*, vol. 15, pp. 545-557, 2020.
- [34] G. Iannaccone, F. Bonaccorso, L. Colombo, and G. Fiori, "Quantum engineering of transistors based on 2D materials heterostructures," *Nature nanotechnology*, vol. 13, pp. 183-191, 2018.
- [35] Y. Liu, X. Duan, H.-J. Shin, S. Park, Y. Huang, and X. Duan, "Promises and prospects of two-dimensional transistors," *Nature*, vol. 591, pp. 43-53, 2021.
- [36] P.-C. Shen, C. Su, Y. Lin, A.-S. Chou, C.-C. Cheng, J.-H. Park, *et al.*, "Ultralow contact resistance between semimetal and monolayer semiconductors," *Nature*, vol. 593, pp. 211-217, 2021/05/01 2021.
- [37] C. D. English, K. K. H. Smithe, R. L. Xu, and E. Pop, "Approaching ballistic transport in monolayer MoS₂ transistors with self-aligned 10 nm top gates," in *2016 IEEE International Electron Devices Meeting (IEDM)*, 2016, pp. 5.6.1-5.6.4.
- [38] K. M. Price, K. E. Schauble, F. A. McGuire, D. B. Farmer, and A. D. Franklin, "Uniform Growth of Sub-5-Nanometer High- κ Dielectrics on MoS₂ Using Plasma-Enhanced Atomic Layer Deposition," *ACS applied materials & interfaces*, vol. 9, pp. 23072-23080, 2017.
- [39] A. Sebastian, R. Pendurthi, T. H. Choudhury, J. M. Redwing, and S. Das, "Benchmarking monolayer MoS₂ and WS₂ field-effect transistors," *Nature Communications*, vol. 12, p. 693, 2021/01/29 2021.
- [40] A. Sebastian, A. Pannone, S. Subbulakshmi Radhakrishnan, and S. Das, "Gaussian synapses for probabilistic neural networks," *Nat Commun*, vol. 10, p. 4199, Sep 13 2019.
- [41] S. Subbulakshmi Radhakrishnan, A. Sebastian, A. Oberoi, S. Das, and S. Das, "A biomimetic neural encoder for spiking neural network," *Nature Communications*, vol. 12, p. 2143, 2021/04/09 2021.
- [42] D. Jayachandran, A. Oberoi, A. Sebastian, T. H. Choudhury, B. Shankar, J. M. Redwing, *et al.*, "A low-power biomimetic collision detector based on an in-memory molybdenum disulfide photodetector," *Nature Electronics*, vol. 3, pp. 646-655, 2020/10/01 2020.
- [43] T. F. Schranghamer, A. Oberoi, and S. Das, "Graphene memristive synapses for high precision neuromorphic computing," *Nature Communications*, vol. 11, p. 5474, 2020/10/29 2020.
- [44] S. Das, A. Dodda, and S. Das, "A biomimetic 2D transistor for audiomorphic computing," *Nature Communications*, vol. 10, p. 3450, 2019/08/01 2019.
- [45] K. Zhu, C. Wen, A. A. Aljarb, F. Xue, X. Xu, V. Tung, *et al.*, "The development of integrated circuits based on two-dimensional materials," *Nature Electronics*, vol. 4, pp. 775-785, 2021/11/01 2021.
- [46] S. Wachter, D. K. Polyushkin, O. Bethge, and T. Mueller, "A microprocessor based on a two-dimensional semiconductor," *Nature communications*, vol. 8, p. 14948, 2017.
- [47] D. K. Polyushkin, S. Wachter, L. Mennel, M. Paur, M. Paliy, G. Iannaccone, *et al.*, "Analogue two-dimensional semiconductor electronics," *Nature Electronics*, vol. 3, pp. 486-491, 2020/08/01 2020.
- [48] D. E. Nikonov and I. A. Young, "Benchmarking of beyond-CMOS exploratory devices for logic integrated circuits," *IEEE Journal on Exploratory Solid-State Computational Devices and Circuits*, vol. 1, pp. 3-11, 2015.

- [49] S. S. Sylvia, K. Alam, and R. K. Lake, "Uniform benchmarking of low-voltage van der Waals FETs," *IEEE Journal on Exploratory Solid-State Computational Devices and Circuits*, vol. 2, pp. 28-35, 2016.
- [50] C.-S. Lee, B. Cline, S. Sinha, G. Yeric, and H. S. P. Wong, "32-bit Processor core at 5-nm technology: Analysis of transistor and interconnect impact on VLSI system performance," pp. 28.3.1-28.3.4, 2016.
- [51] T. Agarwal, A. Szabo, M. G. Bardon, B. Soree, I. Radu, P. Raghavan, *et al.*, "Benchmarking of monolithic 3D integrated MX₂ FETs with Si FinFETs," pp. 5.7.1-5.7.4, 2017.

REVIEWER COMMENTS

Reviewer #2 (Remarks to the Author):

Zheng et al. have improved the paper with the addition of important experimental data and analyses and have satisfied most of my questions. I have a few minor questions/suggestions listed in points 3-5 below that the authors could consider to further improve the paper. Apart from these, my general recommendation is to accept this work for publication in Nature Communications.

1. The added explanation of the origin of cycle-to-cycle variation based on the charge trapping/de-trapping process is appreciated and is a welcome addition to the discussions provided in the paper.
2. The revision of Extended Data Fig. 1 and the associated discussion has clarified the point about hardware scalability with the size of the Bayesian network built out of these devices in the proposed architecture.
3. Regarding the speed of operation of the s-bit generators, the study on pulse width dependence of transfer characteristics up to 100ns presented in Fig. R2 is appreciated. In the same context, the authors mention that the minimum program/erase time that is dependent on the trapping/de-trapping time constants can be as short as several hundred picoseconds. It is requested that the authors provide references in support of this claim in the interest of the reader.
4. The generation of 10^4 random numbers and the associated NIST STS test results (a subset of all the recommended tests) show that the device has the potential to produce high quality random numbers. Although not critical, one question that remains is why not generate more random numbers to complete more tests in the NIST suite, especially since the endurance has been shown to be much higher than 10^4 . Clarifying this would position the potential of the device as an s-bit generator in a clearer sense.
5. The tunability of probability achieved through non-volatile programmability of V_{IT} is not deterministic as has now been clarified by the authors. The associated plots in Fig. R4 on the variation in p_S is appreciated. However, it is not clear if this variation in the programmed probability values will affect the proper operation of the Bayesian network built out of these devices. A discussion on this seems to be important for supporting the claim of these devices being hardware accelerators for Bayesian networks.

Reviewer #3 (Remarks to the Author):

I believe the authors have made proper revisions according to the reviewer's comments. However, I believe that the authors' response regarding the motivation of MoS2 should be included in Introduction.

I think that Introduction is biased too much to Bayesian networks. In addition, I think it would be better to provide how likely to happen for each skewness. For example, skewness=-1 looks like very unlikely to happen, but the readers cannot know how frequently this situation will happen. Perhaps, skewness vs (frequency or probability) plot will be helpful to understand this issue. If the authors can come up with some better plot or data for this concern, it will be also fine.

Reviewer #4 (Remarks to the Author):

In this work, Zheng et al. introduce a hardware “acceleration” of a Bayesian network using 2D-material-based memtransistors. The highlight of the paper is an experimental demonstration with 29 memtransistors. However, the paper has important issues.

1. First, the word “accelerator” should not be used. The scheme proposed by the authors involves programming memtransistors repeatedly (to do stochastic computing), which is a slow operation. Bayesian inference is slower with this accelerator than on a computer. I understand that programming memtransistors might become faster in the future, but it is an inherently slow operation due to memtransistor physics, which CMOS does not require.

2. More profoundly, I am afraid that there is a severe mistake in the energy evaluation of the s-bit circuit (2pJ/bit). This evaluation is not based on measurements, but on Eq. 3, which makes no sense here, and in my opinion, severely underestimates energy consumption for several reasons.

$E=CV^2$ is often used to evaluate the energy consumption of a CMOS logic gate, but this is not a CMOS logic gate. First, there are several stages, so several CV^2 . Second, all stages (voltage divider, inverting amplifier, threshold inverter) are purely NMOS based. During the third phase (when $V_{N1}=V_{DD}$), there will be DC current flowing (this is not as in CMOS, where energy consumption stops as soon as the output has stabilized). If the drain current of the transistors is $\sim 50\mu A$, $V_{DD}\sim 2V$, and the third phase is 100ns (this is my best guess based on the paper), this additional energy consumption is going to be several times 10pJ!

In fact, Eq 3 looks like the energy consumption associated with charging transistor MT1, but it is not the right equation (there is a problem with the 1/2 factor, this is a very classic mistake to just think about the stored energy, and not the whole charge/discharge process).

The authors should measure the real energy consumption of the circuit. I also recommend talking to an electrical engineer with expertise in circuits, and to do, e.g., SPICE simulation.

3. Additionally, I think that the authors exaggerate the energy consumption of s-bit generation in CMOS:

“However, CMOS and FPGA based BN architecture require hundreds of transistors to generate s-bits due, which limits its area and energy efficiency [9].” Ref 9 is an old paper that is very far from the state of the art of low-energy CMOS-based TRNGs.

In rebuttal: "For example, probabilistic CMOS based implementation of BN consumes $2 \sim 150$ pJ energy and requires multiple hardware elements including amplifiers, inverters, flip-flops etc. to generate the s-bits, which increases the transistor count to more than 100 making it energy area inefficient [1]." Ref 1 is even older.

4. Device-to-device variability is a concern for the approach of the authors, and its impact should be investigated.

5. Concerning spintronics, the approach of superparamagnetic tunnel junctions with sense amplifiers should be mentioned and compared to the authors' work as it is highly energy-efficient:

Vodnicarevic, Damir, et al. "Low-energy truly random number generation with superparamagnetic tunnel junctions for unconventional computing." *Physical Review Applied* 8.5 (2017): 054045.

Daniels, Matthew W., et al. "Energy-efficient stochastic computing with superparamagnetic tunnel junctions." *Physical review applied* 13.3 (2020): 034016.

6. Finally, the paper has many imprecisions, e.g.:

- “The success of biological brains in implementing BN lie in the inherently stochastic nature of neural computation ». This is a highly debated hypothesis. A correct phrasing could be “could lie in the...”

- “Certainly, the energy expense can be reduced by reducing the length of the s-bit streams at the cost of reduced precision”. This should be backed by results.

- “The fundamental computing primitive for BN is a s-bit generator” (and similar phrases). This should be “The fundamental computing primitive for THE STOCHASTIC COMPUTING IMPLEMENTATION OF BN” (BN classically do not require s-bits).

- Fig 3f does not have a legend on the x-axis

Reviewer #2 (Remarks to the Author):

Zheng et al. have improved the paper with the addition of important experimental data and analyses and have satisfied most of my questions. I have a few minor questions/suggestions listed in points 3-5 below that the authors could consider to further improve the paper. Apart from these, my general recommendation is to accept this work for publication in Nature Communications.

We are glad that the reviewer is satisfied with our revision. We would also like to thank the reviewer for recommending publication of our work in Nature Communications. We are happy to provide more details regarding the reviewer's remaining questions/suggestions.

1. The added explanation of the origin of cycle-to-cycle variation based on the charge trapping/de-trapping process is appreciated and is a welcome addition to the discussions provided in the paper.

We thank the reviewer for his appreciation of our revision.

2. The revision of Extended Data Fig. 1 and the associated discussion has clarified the point about hardware scalability with the size of the Bayesian network built out of these devices in the proposed architecture.

We thank the reviewer for his appreciation of our revision.

3. Regarding the speed of operation of the s-bit generators, the study on pulse width dependence of transfer characteristics up to 100ns presented in Fig. R2 is appreciated. In the same context, the authors mention that the minimum program/erase time that is dependent on the trapping/de-trapping time constants can be as short as several hundred picoseconds. It is requested that the authors provide references in support of this claim in the interest of the reader.

We agree with the reviewer's suggestion. We have added the following references in the revised manuscript.

[1] H.-S. Tsai, Y.-H. Huang, P.-C. Tsai, Y.-J. Chen, H. Ahn, S.-Y. Lin, et al., "Ultrafast exciton dynamics in scalable monolayer MoS₂ synthesized by metal sulfurization," ACS omega, vol. 5, pp. 10725-10730, 2020.

[2] C. J. Docherty, P. Parkinson, H. J. Joyce, M.-H. Chiu, C.-H. Chen, M.-Y. Lee, et al., "Ultrafast transient terahertz conductivity of monolayer MoS₂ and WSe₂ grown by chemical vapor deposition," ACS nano, vol. 8, pp. 11147-11153, 2014.

4. The generation of 10^4 random numbers and the associated NIST STS test results (a subset of all the recommended tests) show that the device has the potential to produce high quality random numbers. Although not critical, one question that remains is why not generate more random numbers to complete more tests in the NIST suite, especially since the endurance has been shown to be much higher than 10^4 . Clarifying this would position the potential of the device as an s-bit generator in a clearer sense.

We appreciate reviewer's comment. Note that, in order to run the entire set of tests in the NIST suite, we need to generate $> 10^9$ random bits. While possible, it will require unrealistically long time with our present measurement setup due to limitation in the number of data points (~ 1000) that can be collected in a single experiment when the Keysight B1500 semiconductor parameter analyzer is used in the sampling mode. Repeating the experiment 1 million times is a herculean task limiting our current capability to collect $> 10^9$ bits. We are working towards automating the data collection and also communicating with the tool manufacturer to enhance the capability. We can assure the reviewer that in our future demonstrations, we will definitely include larger data sets.

5. The tunability of probability achieved through non-volatile programmability of V_{IT} is not deterministic as has now been clarified by the authors. The associated plots in Fig. R4 on the variation in p_s is appreciated. However, it is not clear if this variation in the programmed probability values will affect the proper operation of the Bayesian network built out of these

devices. A discussion on this seems to be important for supporting the claim of these devices being hardware accelerators for Bayesian networks.

We agree with the reviewer that a discussion on the impact of variation in the programmed probability value on the operation of the Bayesian network (BN) must be included in the revised manuscript. We have now performed numerical simulation to show how the error (ΔP) in the programmed probability impacts the accuracy of the output ($B = AX_1 + A^C X_2$) of the BN. **Fig. R1** shows the colormap of error in $P(B)$ for $P(X_1) = 0.5$, $P(X_2) = 0.56$, and $P(A) = 0.28$. We have used $\Delta P = 0.065$ for both $P(X_1)$ and $P(X_2)$ obtained from **Fig. 3n** of the manuscript. From the error map, it is evident that the variation in the programmed probability inflicted by the cycle-to-cycle programming variation of the memtransistors results in less than 15% error at the output of the BN. We have included these results in the revised *Supplementary Information Fig. S12*.

Figure R1. Simulation results showing percentage error of BN output $P(B)$ for the 2-node BN ($B = AX_1 + A^C X_2$) due to cycle-to-cycle variation of V_{IT} , of the thresholding inverter (**Supplementary Fig. S6b**) leading to uncertainty ($\Delta P \approx 0.065$) in the stochastic input values $P(X_1)$ and $P(X_2)$. $P(X_1)$ represents $P(B/A) = 0.50$, and $P(X_2)$ represents $P(B/A^C) = 0.56$ while the select line, $P(A) = 0.28$, remains constant.

Reviewer #3 (Remarks to the Author):

I believe the authors have made proper revisions according to the reviewer's comments. However, I believe that the authors' response regarding the motivation of MoS₂ should be included in Introduction. I think that Introduction is biased too much to Bayesian networks. In addition, I think it would be better to provide how likely to happen for each skewness. For example, skewness = -1 looks like very unlikely to happen, but the readers cannot know how frequently this situation will happen. Perhaps, skewness vs (frequency or probability) plot will be helpful to understand this issue. If the authors can come up with some better plot or data for this concern, it will be also fine.

We are glad that the reviewer finds our revision satisfactory. We are happy to include more discussion on the motivation behind the use of MoS₂ based memtransistor in the Introduction section. We also agree with the reviewer that skewness = -1 is very unlike to happen. We have now mentioned that in the revised manuscript. We have refrained from including a skewness vs frequency or probability plot as this plot will be mostly hypothetical. We will contemplate if there can be a better plot to convey the impact of skewness in the probability distribution on the performance of the Bayesian network in our future studies.

Reviewer #4 (Remarks to the Author):

In this work, Zheng et al. introduce a hardware “acceleration” of a Bayesian network using 2D-material-based memtransistors. The highlight of the paper is an experimental demonstration with 29 memtransistors. However, the paper has important issues.

We would like to first thank the reviewer for acknowledging our efforts on hardware demonstration of a Bayesian network using 2D-material-based memtransistors. We are happy to resolve the issues the reviewer has mentioned.

1. First, the word “accelerator” should not be used. The scheme proposed by the authors involves programming memtransistors repeatedly (to do stochastic computing), which is a slow operation. Bayesian inference is slower with this accelerator than on a computer. I understand that programming memtransistors might become faster in the future, but it is an inherently slow operation due to memtransistor physics, which CMOS does not require.

We agree with the reviewer’s suggestion. We have replaced the phrase “hardware acceleration” by “hardware implementation” in the title and text in the revised manuscript. We hope that this change is acceptable to the reviewer.

2. More profoundly, I am afraid that there is a severe mistake in the energy evaluation of the s-bit circuit (2pJ/bit). This evaluation is not based on measurements, but on Eq. 3, which makes no sense here, and in my opinion, severely underestimates energy consumption for several reasons. $E=CV^2$ is often used to evaluate the energy consumption of a CMOS logic gate, but this is not a CMOS logic gate. First, there are several stages, so several CV^2 . Second, all stages (voltage divider, inverting amplifier, threshold inverter) are purely NMOS based. During the third phase (when $V_{N1}=V_{DD}$), there will be DC current flowing (this is not as in CMOS, where energy consumption stops as soon as the output has stabilized). If the drain current of the transistors is $\sim 50\mu A$, $V_{DD}\sim 2V$, and the third phase is 100ns (this is my best guess based on the paper), this additional energy consumption is going to be several times 10pJ! In fact, Eq 3 looks like the energy consumption associated with charging transistor MT1, but it is not the right equation (there is a problem with the 1/2 factor, this is a very

classic mistake to just think about the stored energy, and not the whole charge/discharge process). The authors should measure the real energy consumption of the circuit. I also recommend talking to an electrical engineer with expertise in circuits, and to do, e.g., SPICE simulation.

The reviewer's observation is correct and we apologize for our oversight regarding the energy consumption. We have now revised the energy calculation for the s-bit generation circuit using **Eq. R1**.

$$E_{s-bit} = C_G(V_P^2 + V_E^2 + V_R^2 + V_{DD}^2) + \langle I_{N1N4} \rangle V_{DD} \tau_{clk} \quad [\text{R1a}]$$

$$\langle I_{N1N4} \rangle = \frac{1}{n} \sum_{i=1}^n I_{N1N4,i} \quad [\text{R1b}]$$

$$C_G = \epsilon_0 \epsilon_{ox} WL / t_{ox} \quad [\text{R1c}]$$

In **Eq. R1**, V_P , V_E , V_R , and V_{DD} , are the program, erase, read, and supply voltages, respectively, $C_G \approx 10^{-14}$ F is the gate capacitance, $\epsilon_0 = 8.85 \times 10^{-12}$ F/m is the vacuum permittivity, $\epsilon_{ox} = 10$, and $t_{ox} = 50$ nm are, respectively, the relative permittivity and thickness of Al_2O_3 , and $W = 5$ μm and $L = 1$ μm are, respectively, the channel width and length of the 2D-memtransistor. $\langle I_{N1N4} \rangle$ is the average current flowing through the s-bit generator circuit, which is the total current through the voltage divider, inverting amplifier, and threshold inverter during each τ_{clk} . We have used $n = 200$ to calculate the average current per $\tau_{clk} = 100$ μs based on the experimental measurements, as shown in **Fig. R2**. Since most of the memtransistors operate in their respective subthreshold regimes, $\langle I_{N1N4} \rangle \sim 1.5$ nA. As such the second term in **Eq. R1a** accounts for ~ 0.3 pJ, whereas the first term in Eq. R1a accounts for ~ 2 pJ. This results in $E_{s-bit} \sim 2$ pJ/clock-cycle, which supports our claim on energy efficient s-bit generation. We feel that SPICE simulation is beyond the scope of this work and will be included in our future studies.

Figure R2. Total current for the s-bit generation between N_1 and N_4 , i.e., I_{N1N4} for 50 clock cycles.

3. Additionally, I think that the authors exaggerate the energy consumption of s-bit generation in CMOS: “However, CMOS and FPGA based BN architecture require hundreds of transistors to generate s-bits due, which limits its area and energy efficiency [9].” Ref 9 is an old paper that is very far from the state of the art of low-energy CMOS-based TRNGs. In rebuttal: "For example, probabilistic CMOS based implementation of BN consumes 2 ~ 150 pJ energy and requires multiple hardware elements including amplifiers, inverters, flip-flops etc. to generate the s-bits, which increases the transistor count to more than 100 making it energy area inefficient [1].” Ref 1 is even older.

The reviewer’s concern is noted. We have corrected our statements in the revised manuscript accordingly. We have also included new references regarding energy consumption of s-bit generation in CMOS.

[3] Kim, E., Lee, M. & Kim, J. J. in 2017 IEEE International Solid-State Circuits Conference (ISSCC). 144-145.

[4] Pamula, V. R. et al. in 2018 IEEE Symposium on VLSI Circuits. 1-2.

[5] Jayaraj, A., Gujarathi, N. N., Venkatesh, I. & Sanyal, A. 0.6–1.2 V, 0.22 pJ/bit True Random Number Generator Based on SAR ADC. IEEE Transactions on Circuits and Systems II: Express Briefs 67, 1765-1769, doi:10.1109/TCSII.2019.2949775 (2020).

[6] Cao, Y., Zhao, X., Zheng, W., Zheng, Y. & Chang, C. H. A New Energy-Efficient and High Throughput Two-Phase Multi-Bit per Cycle Ring Oscillator-Based True Random Number Generator. IEEE Transactions on Circuits and Systems I: Regular Papers 69, 272-283, doi:10.1109/TCSI.2021.3087512 (2022).

[7] Satpathy, S. et al. in 2018 IEEE Symposium on VLSI Circuits. 169-170.

4. Device-to-device variability is a concern for the approach of the authors, and its impact should be investigated.

The reviewer has raised a valid concern on device-to-device variation. **Fig. R3a** shows the transfer characteristics of 10 MoS₂ memtransistors and **Fig. R3b** shows the transfer characteristics for these 10 devices after one programming/erasing clock cycle ($V_E = 10V$, $V_P = -7V$). Both figures show

small device-to-device variation in our MoS₂ 2D memtransistors, which supports proper operation of hardware Bayesian network with our 2D memtransistors.

Figure R3. (a) Transfer characteristics of 10 MoS₂ memtransistors. (b) Transfer characteristics of the 10 MoS₂ memtransistors after subjecting to programming/erasing voltage pulses ($V_P = -7V$, $V_E = 10V$) each for a duration of $\tau_s = 100 \mu s$.

The impact of device-to-device variation on the operation of the Bayesian network (BN) have been examined through numerical simulation to show how the error (ΔP) in the programmed probability due to device-to-device variation impacts the accuracy of the output ($B = AX_1 + A^C X_2$) of the BN. **Fig. R4** shows the colormap of error in $P(B)$ for $P(X_1) = 0.5$, $P(X_2) = 0.56$, and $P(A) = 0.28$. We have used $\Delta P = 0.046$ for both X_1 and X_2 inferred from **Fig. R3b**. From the error map, it is evident that the variation in the programmed probability inflicted by the device-to-device programming variation of the memtransistors results in a maximum error of 8% at the output of

Figure R4. Simulation results showing percentage error of BN output $P(B)$ for the 2-node BN ($B = AX_1 + A^C X_2$) due to device-to-device variation (**Supplementary Fig. S13b**) leading to uncertainty ($\Delta P \approx 0.046$) in the stochastic input values $P(X_1)$ and $P(X_2)$. $P(X_1)$ represents $P(B/A) = 0.50$, and $P(X_2)$ represents $P(B/A^C) = 0.56$, while the select line, $P(A) = 0.28$, remains constant.

the BN. We have included these data and impact of device-to-device variation in the *Supplementary Information Fig. S13-S14*.

5. Concerning spintronics, the approach of superparamagnetic tunnel junctions with sense amplifiers should be mentioned and compared to the authors' work as it is highly energy-efficient:

Vodenicarevic, Damir, et al. "Low-energy truly random number generation with superparamagnetic tunnel junctions for unconventional computing." *Physical Review Applied* 8.5 (2017): 054045.

Daniels, Matthew W., et al. "Energy-efficient stochastic computing with superparamagnetic tunnel junctions." *Physical review applied* 13.3 (2020): 034016.

We agree with the reviewer's suggestion. The references regarding spintronic approaches based on superparamagnetic tunnel junctions with sense amplifiers is included in the revised manuscript.

6. Finally, the paper has many imprecisions, e.g.:

- "The success of biological brains in implementing BN lie in the inherently stochastic nature of neural computation ». This is a highly debated hypothesis. A correct phrasing could be "could lie in the..."

We agree with the reviewer's suggestion. We have corrected the phrasing.

- "Certainly, the energy expense can be reduced by reducing the length of the s-bit streams at the cost of reduced precision". This should be backed by results.

The reviewer has raised a valid point. We would like to point out the fact that the energy consumption is calculated per s-bit and it was found to be ~ 2 pJ. The bit-length used for BN implementation is 200. While it is possible to encode a given probability using a bitstream of lower bit length, to reduce energy expense, this could lead to a significant loss of precision in the computed BN output for very small bit-lengths. In general, as we reduce the bit-length of s-bit streams involved in BN implementation, we observe a deviation in the probability value encoded

by that bitstream from the actual value. To support this claim, we have performed a numerical simulation to understand the impact of bit-length reduction on the accuracy of the BN output. As expected, the percentage error of the BN output $P(B)$ increases with a reduction in the bit-length used to encode the probability values $P(A) = 0.59$, $P(B/A) = 0.39$, $P(B/A^c) = 0.75$. The

Figure R5. Percentage error of $P(B)$ as a function of bit length of s -bit streams, with the expected value of the BN output, $P(B) = 0.54$, with $P(A) = 0.59$, $P(B/A) = 0.39$, and $P(B/A^c) = 0.75$.

expected BN output in this case is $P(B) = 0.54$. **Fig. R5** shows the percentage error of $P(B)$ as a function of bit length. Clearly, the plot reveals that the percentage error increases with the reduction of the bit-length.

“The fundamental computing primitive for BN is a s -bit generator” (and similar phrases). This should be “The fundamental computing primitive for The stochastic computing implementation of BN” (BN classically do not require s -bits).

The agree with the reviewer’s suggestion. We have corrected the phrasing.

- Fig 3f does not have a legend on the x-axis

Thanks for pointing out the missing level, we have fixed it in the revised manuscript.

References

- [1] H.-S. Tsai *et al.*, "Ultrafast exciton dynamics in scalable monolayer MoS₂ synthesized by metal sulfurization," *ACS omega*, vol. 5, no. 19, pp. 10725-10730, 2020.
- [2] C. J. Docherty *et al.*, "Ultrafast transient terahertz conductivity of monolayer MoS₂ and WSe₂ grown by chemical vapor deposition," *ACS nano*, vol. 8, no. 11, pp. 11147-11153, 2014.
- [3] E. Kim, M. Lee, and J. J. Kim, "8.2 8Mb/s 28Mb/mJ robust true-random-number generator in 65nm CMOS based on differential ring oscillator with feedback resistors," in *2017 IEEE International Solid-State Circuits Conference (ISSCC)*, 5-9 Feb. 2017 2017, pp. 144-145, doi: 10.1109/ISSCC.2017.7870302.
- [4] V. R. Pamula, X. Sun, S. Kim, F. u. Rahman, B. Zhang, and V. S. Sathe, "An All-Digital True-Random-Number Generator with Integrated De-correlation and Bias Correction at 3.2-to-86 MB/S, 2.58 PJ/Bit in 65-NM CMOS," in *2018 IEEE Symposium on VLSI Circuits*, 18-22 June 2018 2018, pp. 1-2, doi: 10.1109/VLSIC.2018.8502375.
- [5] A. Jayaraj, N. N. Gujarathi, I. Venkatesh, and A. Sanyal, "0.6–1.2 V, 0.22 pJ/bit True Random Number Generator Based on SAR ADC," *IEEE Transactions on Circuits and Systems II: Express Briefs*, vol. 67, no. 10, pp. 1765-1769, 2020, doi: 10.1109/TCSII.2019.2949775.
- [6] Y. Cao, X. Zhao, W. Zheng, Y. Zheng, and C. H. Chang, "A New Energy-Efficient and High Throughput Two-Phase Multi-Bit per Cycle Ring Oscillator-Based True Random Number Generator," *IEEE Transactions on Circuits and Systems I: Regular Papers*, vol. 69, no. 1, pp. 272-283, 2022, doi: 10.1109/TCSI.2021.3087512.
- [7] S. Satpathy *et al.*, "An All-Digital Unified Static/Dynamic Entropy Generator Featuring Self-Calibrating Hierarchical Von Neumann Extraction for Secure Privacy-Preserving Mutual Authentication in IoT Mote Platforms," in *2018 IEEE Symposium on VLSI Circuits*, 18-22 June 2018 2018, pp. 169-170, doi: 10.1109/VLSIC.2018.8502369.

REVIEWERS' COMMENTS

Reviewer #2 (Remarks to the Author):

The authors have satisfactorily addressed all of my comments. I am happy to recommend this manuscript for publication in Nature Communications.

Reviewer #3 (Remarks to the Author):

I appreciate the authors efforts to address the reviewers comments, and I believe that this manuscript is ready to be published.

Reviewer #4 (Remarks to the Author):

The authors have addressed my concerns.

I think that the energy estimate is still very rough. It is probably acceptable, but the text could be edited to embrace that this is just a rough estimate (e.g., "The average energy ... was calculated..." could be "A rough estimate of the energy... can be calculated...").

REVIEWERS' COMMENTS

Reviewer #2 (Remarks to the Author):

The authors have satisfactorily addressed all of my comments. I am happy to recommend this manuscript for publication in Nature Communications.

We are happy to learn that the reviewer is satisfied with our response and revision. We would also like to thank the reviewer for recommending the publication of our manuscript in Nature Communication.

Reviewer #3 (Remarks to the Author):

I appreciate the authors efforts to address the reviewers' comments, and I believe that this manuscript is ready to be published.

We are happy to learn that the reviewer is satisfied with our response and revision. We would also like to thank the reviewer for recommending the publication of our manuscript in Nature Communication.

Reviewer #4 (Remarks to the Author):

The authors have addressed my concerns.

I think that the energy estimate is still very rough. It is probably acceptable, but the text could be edited to embrace that this is just a rough estimate (e.g., "The average energy ... was calculated..." could be "A rough estimate of the energy... can be calculated...").

We are happy to learn that the reviewer is satisfied with our response and revision. We would also like to thank the reviewer for recommending the publication of our manuscript in Nature Communication. We agree with the reviewer's suggestion. We have revised the text accordingly.